# Multi-protein bridging factor 1(Mbf1), Rps3 and Asc1 prevent stalled ribosomes from frameshifting

Jiyu Wang[1,2], Jie Zhou[1], Qidi Yang[1], Elizabeth J Grayhack[1,2]*

[1]Department of Biochemistry and Biophysics, School of Medicine and Dentistry, University of Rochester, Rochester, New York; [2]Center for RNA Biology, University of Rochester, Rochester, New York

**Abstract** Reading frame maintenance is critical for accurate translation. We show that the conserved eukaryotic/archaeal protein Mbf1 acts with ribosomal proteins Rps3/uS3 and eukaryotic Asc1/RACK1 to prevent frameshifting at inhibitory CGA-CGA codon pairs in the yeast *Saccharomyces cerevisiae*. Mutations in *RPS3* that allow frameshifting implicate eukaryotic conserved residues near the mRNA entry site. Mbf1 and Rps3 cooperate to maintain the reading frame of stalled ribosomes, while Asc1 also mediates distinct events that result in recruitment of the ribosome quality control complex and mRNA decay. Frameshifting occurs through a +1 shift with a CGA codon in the P site and involves competition between codons entering the A site, implying that the wobble interaction of the P site codon destabilizes translation elongation. Thus, eukaryotes have evolved unique mechanisms involving both a universally conserved ribosome component and two eukaryotic-specific proteins to maintain the reading frame at ribosome stalls.
DOI: https://doi.org/10.7554/eLife.39637.001

*For correspondence:
elizabeth_grayhack@urmc.
rochester.edu

Competing interests: The authors declare that no competing interests exist.

## Introduction

Accurate translation of mRNA into protein depends upon precise, repetitive three base translocation of the ribosome to maintain the correct reading frame throughout a coding sequence. Reading frame maintenance is challenging because multiple movements of the tRNAs and mRNA as well as conformational changes within the ribosome itself are required to complete a single elongation cycle (*Noller et al., 2017*). For instance, the tRNA acceptor stems move within the large subunit during formation of the hybrid state, while the joining of EF-G-GTP (eEF2 in eukaryotes) results in additional movement of tRNA, and finally completion of translocation, driven by Pi release, requires additional movements (*Belardinelli et al., 2016*; *Brilot et al., 2013*; *Noller et al., 2017*; *Pulk and Cate, 2013*; *Ramrath et al., 2013*; *Ratje et al., 2010*; *Tourigny et al., 2013*; *Zhou et al., 2014*). To accomplish this cycle, many interactions between the tRNAs and ribosome are disrupted, and new interactions are created, but the relative position of the tRNA anticodon to the mRNA codon must be maintained throughout all of these events (*Noller et al., 2017*; *Dever et al., 2018*; *Rodnina, 2018*). Thus, it is critical that mechanisms exist to prevent slippage during these transitions.

Reading frame maintenance is facilitated by structures within the ribosome as well as by tRNA modifications. Structural features that contribute to reading frame maintenance, inferred from analysis of prokaryotic translation intermediates, include a swivel of the 30S head relative to the 30S body to form a contracted mRNA tunnel downstream of the A site prior to translocation (*Jenner et al., 2010*; *Schuwirth et al., 2005*). In addition, during translocation, two conserved bases in the 16S rRNA intercalate into different positions of the mRNA to prevent slippage (*Zhou et al., 2013*), while domain IV of EF-G contacts the codon and tRNA in the A/P site and h44 of 16S rRNA, likely coupling mRNA and tRNA movement (*Ramrath et al., 2013*; *Zhou et al., 2014*). tRNA modifications within

**eLife digest** Proteins perform all the chemical reactions needed to keep a cell alive; thus, it is essential to assemble them correctly. They are made by molecular machines called ribosomes, which follow a sequence of instructions written in genetic code in molecules known as mRNAs. Ribosomes essentially read the genetic code three letters at a time; each triplet either codes for the insertion of one of 20 building blocks into the emerging protein, or serves as a signal to stop the process. It is critical that, after reading one triplet, the ribosome moves precisely three letters to read the next triplet. If, for example, the ribosome shifted just two letters instead of three – a phenomenon known as "frameshifting" – it would completely change the building blocks that were used to make the protein. This could lead to atypical or aberrant proteins that either do not work or are even toxic to the cell.

For a variety of reasons, ribosomes will often stall before they have finished building a protein. When this happens, the ribosome is more likely to frameshift. Cells commonly respond to stalled ribosomes by recruiting other molecules that work as quality control systems, some of which can disassemble the ribosome and break down the mRNA. In budding yeast, one part of the ribosome – named Asc1 – plays a key role in recruiting these quality control systems and in mRNA breakdown. If this component is removed, stalled ribosomes frameshift more frequently and, as a result, aberrant proteins accumulate in the cell. Since the Asc1 recruiter protein sits on the outside of the ribosome, it seemed likely that it might act through other factors to stop the ribosome from frameshifting when it stalls. However, it was unknown if such factors exist, what they are, or how they might work.

Now, Wang et al. have identified two additional yeast proteins, named Mbf1 and Rps3, which cooperate to stop the ribosome from frameshifting after it stalls. Rps3, like Asc1, is a component of the ribosome, while Mbf1 is not. It appears that Rps3 likely stops frameshifting via an interaction with the incoming mRNA, because a region of Rps3 near the mRNA entry site to the ribosome is important for its activity. Further experiments then showed that the known Asc1-mediated breakdown of mRNAs did not depend on Mbf1 and Rps3, but also assists in stopping frameshifting. Thus, frameshifting of stalled ribosomes is prevented via two distinct ways: one that directly involves Mbf1 and Rps3 and one that is promoted by Asc1, which reduces the amounts of mRNAs on which ribosomes frameshift.

These newly identified factors may provide insights into the precisely controlled protein-production machinery in the cell and into roles of the quality control systems. An improved understanding of mechanisms that prevent frameshifting could eventually lead to better treatments for some human diseases that result when these processes go awry, which include certain neurological conditions.

DOI: https://doi.org/10.7554/eLife.39637.002

the anticodon loop also assist in reading frame maintenance, inferred both from genetic and structural analyses. Mutants that affect several such modifications in both bacteria and eukaryotes result in increased frameshifting (*Atkins and Björk, 2009*; *Jäger et al., 2013*; *Tükenmez et al., 2015*; *Urbonavicius et al., 2001*; *Waas et al., 2007*). Moreover, a cross-strand base stacking interaction between a modified ms$^2$i$^6$A37 in an *E. coli* tRNA$^{Phe}$ and the mRNA codon is proposed to prevent slippage of P site tRNA on the mRNA (*Jenner et al., 2010*). Thus, a number of mechanisms exist to prevent loss of reading frame.

Nevertheless, ribosomes do move into alternative reading frames in response to specific sequences and structures in mRNA (*Atkins and Björk, 2009*; *Dever et al., 2018*; *Dinman, 2012*). The existence of such events has implied that ribosomal plasticity with respect to reading frame movement is an integral function of the translation machinery. The common feature of all frameshifting events in bacteria to humans is that the ribosome stalls (*Dever et al., 2018*). The stall can be mediated by combined effects of the A and P site codons (*Farabaugh et al., 2006*; *Gamble et al., 2016*), by the presence of downstream structures, or by an upstream Shine-Dalgarno sequence in bacteria (*Caliskan et al., 2014*; *Dinman, 2012*). Analysis of programmed frameshifting indicates that additional sequences or protein factors are frequently required to mediate efficient frameshifting (*Atkins and Björk, 2009*; *Dinman, 2012*). For instance, +1 programmed frameshifting events are

frequently enhanced by stimulatory sequences, although the role of these sequences is not always clear (*Guarraia et al., 2007*; *Taliaferro and Farabaugh, 2007*).

The identification of mutants that either affect programmed frameshifting or suppress frameshift mutations has pointed to four key factors in reading frame maintenance. First, mutations of ribosomal proteins, particularly those that contact the P site tRNA can cause increased frameshifting. In bacteria, frameshifting mutations are suppressed by deletions within the C terminal domain of ribosomal protein uS9, which contacts the P site tRNA anticodon loop (*Jäger et al., 2013*). In the yeast *Saccharomyces cerevisiae*, programmed frameshifting in the L-A virus is affected by mutations in 5S rRNA or its interactors uL18 or uL5, that also contact the P site tRNA (*Meskauskas and Dinman, 2001*; *Rhodin and Dinman, 2010*; *Smith et al., 2001*). Frameshifting mutations are also suppressed by a mutation in the yeast *RPS3*, although this mutation does not affect a tRNA contact (*Hendrick et al., 2001*). Second, mutations in the basal translation machinery can also affect frameshifting. For instance, frameshifting mutations are suppressed by mutations in both EF-1α, which delivers tRNA to the ribosome (*Sandbaken and Culbertson, 1988*), and in *SUP35*, encoding the translation termination factor eRF3 (*Wilson and Culbertson, 1988*). Third, miRNAs can affect the efficiency of programmed frameshifting, for instance at *CCR5* in humans (*Belew et al., 2014*). Fourth, mutations that affect proteins with previously unknown functions in translation can either alter programmed frameshifting or suppress frameshifting mutations. For instance, in yeast, frameshifting mutations are suppressed by mutations in *MBF1*, encoding Multi-protein Bridging Factor 1 (*Hendrick et al., 2001*), or in *EBS1* (*Ford et al., 2006*), while in the porcine virus PRRSV, the RNA binding protein nsp1β stimulates both −1 and −2 frameshifting events (*Li et al., 2014*). Thus, reading frame maintenance is modulated by ribosomal components, many of which contact the tRNAs, as well as by non-ribosomal proteins and miRNAs. However, the roles of many of these proteins are not understood.

We set out to work out the mechanisms that maintain reading frame when eukaryotic ribosomes encounter a stall, the common feature of all frameshifting events. In bacteria, ribosome stalls due to limited availability or functionality of tRNA seem to suffice to cause frameshifting (*Gurvich et al., 2005*; *Seidman et al., 2011*). In wild type yeast, ribosomes stall at CGA codon repeats, which inhibit translation due to wobble decoding of CGA by its native tRNA$^{Arg(ICG)}$ (*Letzring et al., 2010*; *Letzring et al., 2013*). Although frameshifting was detected at several underrepresented heptanucleotide sequences in yeast (*Shah et al., 2002*), it was not detected at CGA codon repeats (*Wolf and Grayhack, 2015*). Instead, eukaryotes have evolved new pathways to regulate inefficient translation events, such as the Ribosome Quality Control (RQC) pathway, in which these stalled ribosomes undergo ubiquitination of ribosomal proteins, followed by dissociation of the subunits, and recruitment of the RQC Complex, which mediates CAT tailing and degradation of the nascent polypeptide (*Bengtson and Joazeiro, 2010*; *Brandman and Hegde, 2016*; *Brandman et al., 2012*; *Defenouillère et al., 2013*; *Joazeiro, 2017*; *Kostova et al., 2017*; *Shao et al., 2013*; *Shen et al., 2015*; *Verma et al., 2013*; *Wilson et al., 2007*; *Juszkiewicz and Hegde, 2017*; *Matsuo et al., 2017*; *Simms et al., 2017b*; *Sitron et al., 2017*; *Sundaramoorthy et al., 2017*). The ribosomal protein Asc1/RACK1 mediates these events (*Brandman et al., 2012*; *Kuroha et al., 2010*); in the absence of Asc1, ribosomes fail to engage the RQC (*Sitron et al., 2017*), and also undergo substantial frameshifting at CGA codon repeats (*Wolf and Grayhack, 2015*). However, Asc1 sits on the outside of the ribosome at the mRNA exit tunnel and likely functions as scaffold for recruitment of other proteins, such as the E3 ubiquitin ligase Hel2/mammalian ZNF598 and Slh1 (*Juszkiewicz and Hegde, 2017*; *Matsuo et al., 2017*; *Simms et al., 2017b*; *Sitron et al., 2017*; *Sundaramoorthy et al., 2017*). Based on the location of Asc1 and the precedent that Asc1 recruits other proteins to abort translation, we considered it likely that Asc1 cooperates with additional proteins to mediate reading frame maintenance at CGA codon repeats and set out to find such factors.

Here, we provide evidence that the Multi-protein Bridging Factor 1 (Mbf1) and ribosomal proteins Rps3 and Asc1 (homolog of human Rack1) work together to prevent translational slippage at CGA codon repeats. Frameshifting results from inactivation of *MBF1*, or from mutations in amino acids in Rps3 located on an exposed surface of the protein near the mRNA entry site. Asc1 was previously known to mediate recruitment of the RQC, mRNA cleavage and mRNA decay at similar stall sites (*Ikeuchi and Inada, 2016*; *Kuroha et al., 2010*; *Letzring et al., 2013*; *Sitron et al., 2017*), as well as reading frame maintenance (*Wolf and Grayhack, 2015*). We provide evidence that Asc1 and Mbf1 cooperate to mediate reading frame maintenance at similar sites, acting on a common set of

substrates, including the seven most slowly translated codon pairs in yeast (*Gamble et al., 2016*). We examined the precise frameshift at one of these inhibitory pairs, CGA-CGG, purifying the frame-shifted polypeptide, followed by analysis with mass spectrometry. We found that frameshifting occurs in the +1 direction at the CGA codon and moreover, that frameshifting is modulated by the competition between the in-frame and +1 frame tRNAs.

## Results

### *MBF1* (Multi-protein Bridging Factor 1) prevents frameshifting at CGA codon repeats

We considered it likely that proteins other than Asc1 worked to prevent frameshifting at CGA codon repeats for two reasons. First, Asc1 binds on the outside of the ribosome, not in the decoding center (*Rabl et al., 2011*), and thus is not positioned in any obvious way to assist with reading frame maintenance. Second, Asc1 recruits other proteins, Hel2 and Slh1, to recruit the RQC (*Brandman and Hegde, 2016*; *Joazeiro, 2017*; *Sitron et al., 2017*), and thus is likely to work with other proteins in reading frame maintenance. Thus, we set out to identify genes responsible for reading frame maintenance at CGA codon repeats.

To isolate mutants that frameshift due to translation of CGA codon repeats, we set up a selection in which a +1 frameshift (caused by 6 adjacent CGA codons) was required to express the *URA3* gene. The native *URA3* gene was placed in the +1 reading frame downstream of an N-terminal domain of *GLN4* encoding amino acids 1–99 (*GLN4*$_{(1-99)}$), followed by 6 CGA codons and one additional nucleotide upstream of the *URA3* coding region (*Figure 1A*). This strain exhibits an Ura⁻ phenotype, due to the low levels of frameshifting in an otherwise wild-type background, allowing for an Ura⁺ selection to obtain mutants with increased frameshifting. As an independent secondary screen for frameshifting mutants due to CGA codon repeats, we integrated a modified version of the RNA-ID reporter with *GLN4*$_{(1-99)}$ followed by 4 CGA codons and one additional nucleotide upstream of the *GFP* coding region into the *ADE2* locus (*Dean and Grayhack, 2012*; *Wolf and Grayhack, 2015*). Thus, GFP expression was dependent upon frameshifting efficiency (*Figure 1A*). To avoid obtaining mutations in the *ASC1* gene, the selection strain also contained a second copy of the *ASC1* gene on a plasmid. (*Figure 1A*). We selected Ura⁺ mutants from 40 independent cultures each of *MATa* and *MATα* parents and then analyzed three Ura⁺ mutants from each culture by flow cytometry to measure GFP and RFP expression. Most mutants (60% of *MATα* mutants and 80% of *MATa* mutants) showed elevated expression of GFP, and we studied those that exhibited relatively high levels of frameshifting, >30% of that in an *asc1Δ* mutant (*Figure 1B*). Most mutants (43 of 48 examined) were recessive and mapped to a single complementation group, based on growth of diploids on media lacking uracil (*Figure 1—figure supplement 1A*), although four dominant mutants were also identified.

To confirm that inhibitory decoding of CGA codon repeats is required for frameshifting in these mutants, we showed that introduction of an anticodon-mutated exact match tRNA$^{Arg(UCG)*}$ suppressed the Ura⁺ phenotype of one mutant (*Figure 1C*). We have shown previously that expression of this exact match tRNA$^{Arg(UCG)*}$ results in efficient decoding of CGA codons and suppresses their inhibitory effects on gene expression (*Letzring et al., 2010*). Thus, the Ura⁺, GFP⁺ phenotype of this mutant was due to frameshifting that occurs when the ribosome translates CGA codon repeats inefficiently.

We demonstrated that mutations in the yeast gene *MBF1*, Multi-protein Bridging Factor 1, were responsible for the defects in reading frame maintenance in recessive high GFP mutants. We identified the mutated gene by complementation of the Ura⁺ phenotype of the P25 recessive mutant with two plasmids from a library that contains 97.2% of the entire yeast genome (*Figure 1—figure supplement 1B*) (*Jones et al., 2008*). The complementing plasmids share a single ORF, *MBF1*. We confirmed that mutations in the *MBF1* gene are responsible for frameshifting in three ways. First, a plasmid with only the *MBF1* gene complemented the frameshifting Ura⁺ phenotype of two mutants (*Figure 1—figure supplement 2A*). Second, deletion of *MBF1* in the parent selection strain (*Figure 1A*) converted that strain from GFP⁻ to GFP⁺, similar to deletion of *ASC1* (*Figure 1—figure supplement 2B*). Third, 19/19 mutants tested contain mutations in the *MBF1* gene. Point mutations

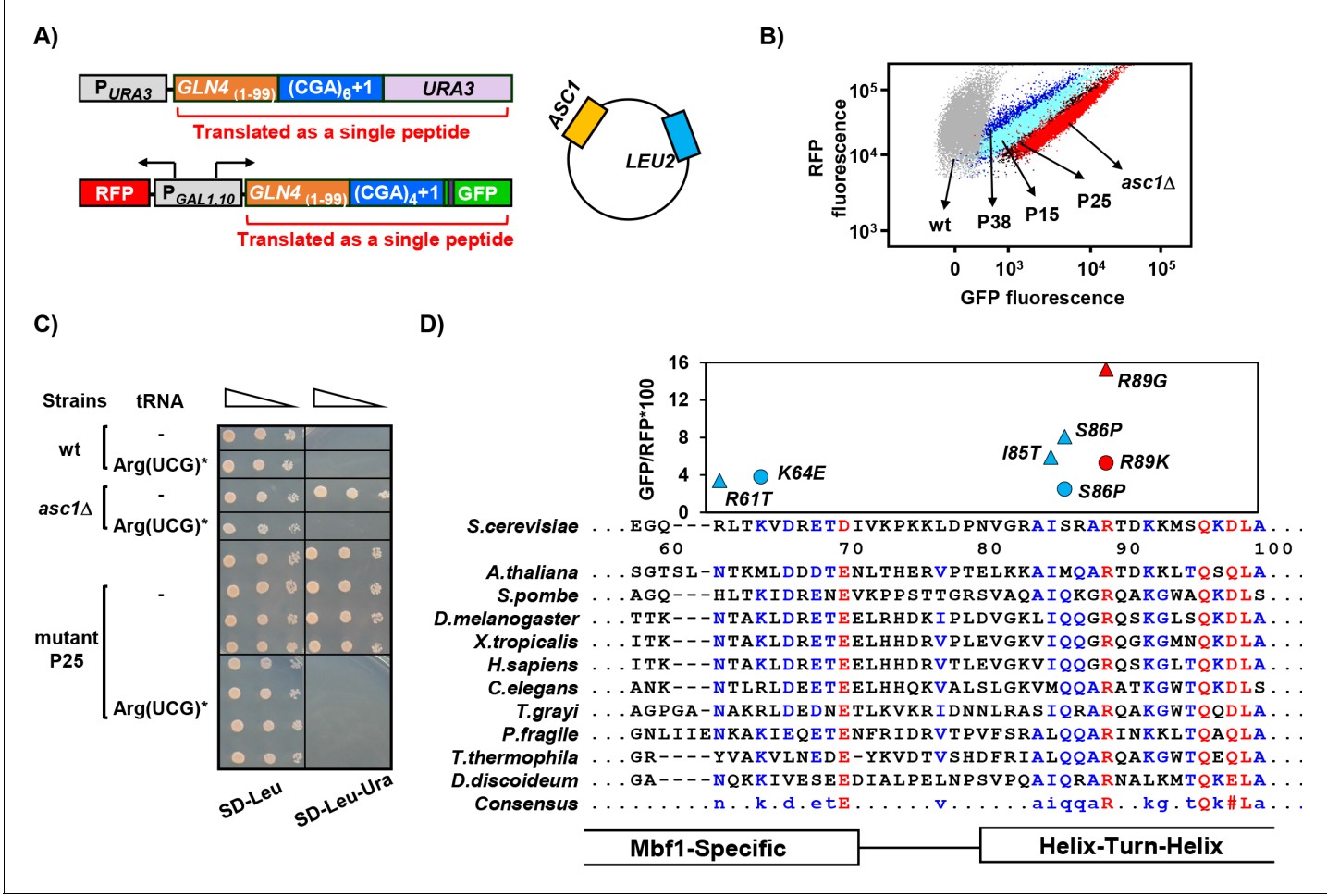

**Figure 1.** *MBF1* (Multi-protein Bridging Factor 1) prevents frameshifting at CGA codon repeats. (**A**) Schematic of selection for mutants that frameshift at CGA codon repeats. The indicated CGA codon repeats plus one extra nucleotide were inserted upstream of the *URA3* and *GFP* coding region (with an upstream HA tag shown in purple), resulting in an Ura⁻ GFP⁻ parent strain. Additional copies of the *ASC1* gene were introduced on a *LEU2* plasmid to avoid recessive mutations in the native *ASC1* gene. To obtain mutants with increased frameshifting efficiency, Ura⁺ mutants were selected and screened for increased GFP/RFP. (**B**) Expression of $GLN4_{(1-99)}$-(CGA)$_4$+1-GFP is increased in the *MATa* Ura⁺ mutants from this selection. Flow cytometry scatter plot showing GFP fluorescence versus RFP fluorescence for three mutants from this selection (P15: *mbf1-R89K*, light blue; P25: *mbf1Δ125–151*, black; P38: *mbf1-K64E*, dark blue), for the *asc1Δ* mutant (red) and for the wild-type parent strain (gray). (**C**) Expression of the non-native tRNA$^{Arg(UCG)*}$ suppressed the Ura⁺ phenotype of mutant P25 at 30°C. Serial dilutions of the indicated strains with empty vector or expressing the mutant tRNA$^{Arg(UCG)*}$ were grown on the indicated media. (**D**) Mutations in the *MBF1* mutants map in conserved amino acids in both the *MBF1*-specific domain and the Helix-Turn-Helix (HTH) domain of Mbf1 protein. Alignment of yeast Mbf1 amino acids 60–100 with other eukaryotic species is shown (full alignment see *Figure 1—figure supplement 3A*). GFP/RFP of frameshifted (CGA)$_4$+1 reporter is shown for mutants obtained from *MATa* (circles) and *MATα* (triangles) strains, with the color of markers corresponding to the consensus level of this residue (Blue: 50–90%, Red: >90%), however the conserved residue for R61 is N, and for S86 is Q, with all others identical to yeast.

DOI: https://doi.org/10.7554/eLife.39637.003

The following figure supplements are available for figure 1:

**Figure supplement 1.** Classification of dominant and recessive mutations and complementation of a recessive mutation.
DOI: https://doi.org/10.7554/eLife.39637.004

**Figure supplement 2.** Confirmation that mutations in *MBF1* are responsible for frameshifting.
DOI: https://doi.org/10.7554/eLife.39637.005

**Figure supplement 3.** Mbf1 is conserved and frameshifting mutations do not exhibit sensitivity to 3-AT.
DOI: https://doi.org/10.7554/eLife.39637.006

isolated in our selection are located at conserved residues near the junctions between two domains (*Figure 1D*).

*MBF1* is a highly conserved gene in eukaryotes and archaea, generally less than 160 amino acids with an N-terminal Mbf1-specific domain (that differs between archaea and eukaryotes) and a conserved cro-like helix-turn-helix (HTH) domain (*Figure 1D*, *Figure 1—figure supplement 3A*). Mbf1, which was initially identified as a transcription co-activator in *Bombyx mori* (*Li et al., 1994*; *Takemaru et al., 1997*), has been implicated in a similar function in yeast, in this case, interacting with the Gcn4, transcription regulator of the general amino acid control pathway (*Takemaru et al., 1998*). In testing sensitivity to 3-aminotriazole (3-AT) (*Hilton et al., 1965*; *Schürch et al., 1974*), a phenotype of *gcn4* mutants due to inability to induce expression of *HIS3*, we found that two frameshifting point mutants (*mbf1-K64E* and *mbf1-I85T*) exhibit no growth defect even on high concentrations of 3-AT (*Figure 1—figure supplement 3B*). Moreover, deletion of *GCN4* does not affect frameshifting at CGA codon repeats in an *asc1Δ* mutant (*Wolf and Grayhack, 2015*). Thus, it is unlikely that the defect in reading frame maintenance in our *mbf1* mutants is related to *GCN4*. However, Mbf1 has also been implicated in translation, based on isolation of mutations in yeast *MBF1* that suppress frameshifting mutations (*Hendrick et al., 2001*), and the weak association of the archaeal homolog with ribosomes (*Blombach et al., 2014*), but there is no information on its molecular role in translation.

## Ribosomal protein Rps3 also mediates reading frame maintenance at CGA codon repeats

To identify the mutated gene(s) in our dominant mutants, we performed whole genome sequencing in two *MATα* mutants and found that each mutant contains a single amino acid change (*S104Y* and *G121D*) in *RPS3*. Similarly, the two dominant *MATa* mutants also contain mutations in the *RPS3* gene (*L113F* and a duplication of N22 to A30). *RPS3* encodes a universally conserved ribosomal protein, a core component of the mRNA entry tunnel with a eukaryotic-specific C-terminal extension that interacts with Asc1 (*Rabl et al., 2011*). One known mutation in *RPS3* (*K108E*) affects reading frame maintenance (*Hendrick et al., 2001*), while others affect different aspects of translation, from initiation to quality control (*Dong et al., 2017*; *Graifer et al., 2014*; *Limoncelli et al., 2017*; *Takyar et al., 2005*). The three residues *S104*, *L113* and *G121* implicated in reading frame maintenance in our study, as well as *K108*, are all found in two α-helices near the mRNA entry tunnel of the ribosome; these residues reside on the surface of the ribosome and could interact with mRNA or proteins outside of the ribosome (*Figure 2A*). Moreover, the identity of all four of these residues is conserved in eukaryotes, but different in bacteria and archaea (*Graifer et al., 2014*).

We initially examined the effect of the *RPS3-K108E* mutation on both frameshifting and in-frame expression downstream of CGA codon repeats, and found that this mutation allows frameshifting but does not affect in-frame expression. We chose the *K108E* mutation because it is known to have only minor effects on the polysome to monosome ratio (*Dong et al., 2017*), consistent with few nonspecific effects on translation. We introduced modified RNA-ID reporters into *rps3Δ::ble^R* strains in which the only source of *RPS3* is a plasmid-borne copy (either wild type or *K108E*). As described previously, since the expression of GFP and RFP is driven by the bi-directional *GAL1,10* promoter, we use the ratio of GFP/RFP to reduce noise and cell type specific differences in induction of this promoter (*Dean and Grayhack, 2012*). Neither the *RPS3* mutant nor a *mbf1Δ* mutant had a substantial effect on GFP/RFP fluorescence (protein), mRNA or protein/mRNA of reporters with CGA or AGA codon repeats in-frame (*Figure 2B*). We did note relatively minor, but compensatory effects, of the mutants on both GFP and RFP mRNAs (a 15–30% reduction in *mbf1Δ* mutants and a similar increase in the *RPS3–K108E* mutant) (*Figure 2—figure supplement 1A*). The *RPS3-K108E* and *mbf1Δ* mutants each caused substantially increased frameshifted GFP/RFP protein and protein/mRNA in the construct with four CGA codons, but had only small effects on GFP/RFP mRNA; no frameshifting was seen with four AGA codons (*Figure 2B*; *Supplementary file 1*).

If Mbf1 and Rps3 proteins function in independent pathways to prevent frameshifting, we expected that *RPS3-K108E mbf1Δ* double mutants would frameshift more efficiently than either single mutant. Instead, we found that the double mutant *RPS3-K108E mbf1Δ* exhibited only a slight increase in frameshifted GFP/RFP protein; this increase is likely due to a slight increase in GFP/RFP mRNA relative to either single mutant, resulting in nearly identical protein/mRNA from the *mbf1Δ* and *RPS3-K108E mbf1Δ* mutants (*Figure 2B*, *Figure 2—figure supplement 1B*). We also examined

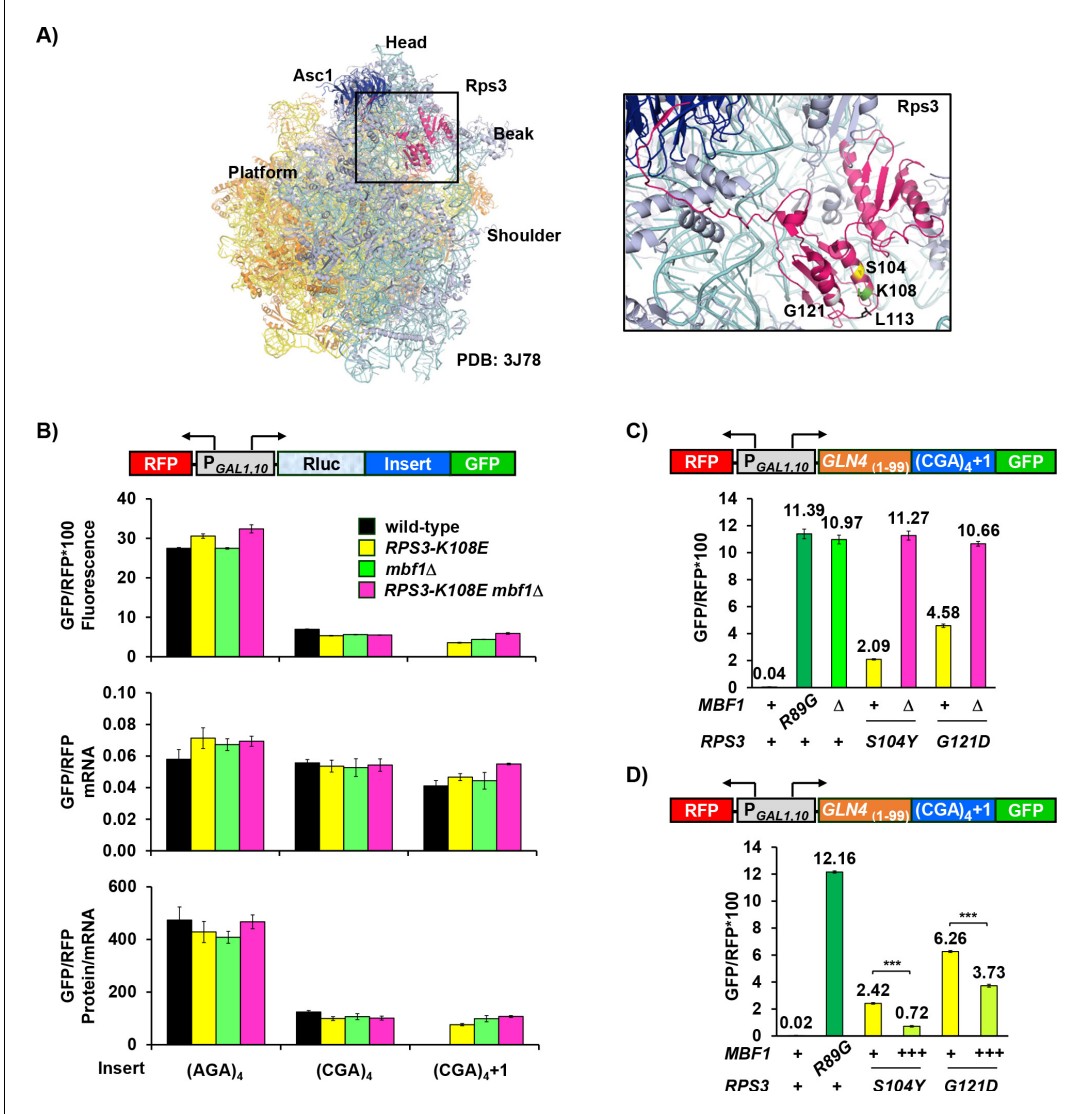

**Figure 2.** Ribosomal protein Rps3 mediates reading frame maintenance with Mbf1 at CGA codon repeats. (A) Left: Yeast ribosome from PDB: 3J78 (*Svidritskiy et al., 2014*) (light blue: small subunit; yellow: large subunit) showing Asc1/RACK1 (dark blue) and Rps3 (pink). Right: Residues of Rps3 in which mutations cause frameshifting are marked- *S104* (yellow), *K108* (green), *L113* (black), *G121* (gray). (B) Analysis of effects of *RPS3-K108E, mbf1Δ and RPS3-K108E mbf1Δ* mutations on expression of GFP/RFP protein (fluorescence), mRNA and protein/mRNA from reporters containing four Arg codons (AGA versus CGA) in-frame and in the +1 frame. (C) Assay for epistatic relationship between *RPS3* mutations from this selection and the *mbf1Δ* mutation. (D) Overproduction of Mbf1 protein in indicated *RPS3* mutants significantly decreased expression of frameshifted Gln4(1-99)-GFP fusion protein (***p < 0.001) analyzed by flow cytometry.

DOI: https://doi.org/10.7554/eLife.39637.007

The following source data and figure supplement are available for figure 2:

**Source data 1.** (Source data file for *Figure 2B–D*).
DOI: https://doi.org/10.7554/eLife.39637.009
**Figure supplement 1.** *RPS3* and *mbf1Δ* mutants do not affect mRNA levels of CGA-containing reporters.
DOI: https://doi.org/10.7554/eLife.39637.008

effects of combining *MBF1* mutants with other *RPS3* mutants, *S104Y* and *G121D* from our selection, to determine epistasis. In these cases again, each single mutant exhibited frameshifting and the double mutants exhibited similar amounts of frameshifted GFP/RFP to that in the *mbf1Δ* strain, although even an additive effect would be easily detectable (*Figure 2C*). Thus, we think it is likely that Mbf1

and the two α-helices in the N-terminal Rps3 protein have related roles in reading frame maintenance.

If Mbf1 and these two α-helices in Rps3 mediate a common function, then frameshifting in either *RPS3-S104Y* or *G121D* mutants might be suppressed by overproduction of *MBF1*. We found that introduction of additional copies of the *MBF1* gene into either of these mutants resulted in reduced expression of frameshifted GFP (*Figure 2D*). Frameshifted GFP was reduced to 30% in the *S104Y* mutant and to 60% in the *G121D* mutant (*Figure 2D*). Similarly, growth on media lacking uracil was severely compromised in the *RPS3-S104Y* mutant when *MBF1* was expressed on a multi-copy plasmid, relative to an empty vector control (*Figure 2—figure supplement 1C*), although both strains grow equally well on SD-Leu media. These observations are consistent with the idea that Mbf1 and Rps3 play similar roles in reading frame maintenance and support the idea that these *RPS3* mutations reduce Mbf1 function.

## Mbf1 and Asc1 play distinct, but related, roles at CGA codon repeats

Since Asc1 is also required for reading frame maintenance at CGA codon repeats (*Wolf and Grayhack, 2015*), we examined the relationship between *MBF1* and *ASC1* on both frameshifting and in-frame expression, comparing GFP/RFP in the single mutants to that in the *asc1Δ mbf1Δ* double mutant. Asc1 is known to affect expression (both in-frame and +1 frame) downstream of four CGA codons (*Letzring et al., 2013*; *Wolf and Grayhack, 2015*), but we previously noted that inhibitory effects of CGA codons are mediated by CGA codon pairs (*Gamble et al., 2016*; *Letzring et al., 2010*). Therefore, we compared the effects of these mutants on a set of reporters with three CGA-CGA (or AGA-AGA) codon pairs flanked by two non-Arg codons (*Figure 3A*, *Figure 3—figure supplement 1A*, *Supplementary file 2*) to effects on a set with four adjacent CGA (or AGA) codons (*Figure 3—figure supplement 1B*). Neither the upstream gene nor the arrangement of CGA codons affected the results.

As expected based on a previous report (*Sitron et al., 2017*), deletion of *ASC1* resulted in increased protein and mRNA levels of reporters with in-frame CGA-CGA codon pairs. By contrast, deletion of *MBF1* did not affect protein or mRNA levels substantially (*Figure 3A*, *Figure 3—figure supplement 1A and B*, *Supplementary file 2*). Thus, Asc1 clearly has a unique role in regulating mRNA and RQC recruitment at CGA codon pairs, but overall expression, measured as protein/mRNA, of all in-frame reporters is similar in the wild type, *asc1Δ*, *mbf1Δ*, and *asc1Δ mbf1Δ* double mutants (*Figure 3A*). For the in-frame reporters, the relationship between GFP/RFP fluorescence and mRNA is linear; none of these mutants affect RFP mRNA (*Figure 3—figure supplement 1C and D*). However, the increase in mRNA in an *asc1Δ* mutant does not explain the increase in frameshifted GFP/RFP protein in this mutant. That is, the 2.7-fold increase in GFP/RFP mRNA from the +1 reporter in an *asc1Δ* mutant (relative to the wild type) cannot account for the >50 fold increase in frameshifted GFP/RFP fluorescence (*Figure 3A*). Thus, an *asc1Δ* mutant clearly exhibits a defect in reading frame maintenance.

If increased frameshifted protein/mRNA in an *asc1Δ* mutant is due to a failure of the Mbf1 pathway, then we expected that *asc1Δ mbf1Δ* double mutants would frameshift with similar efficiency to the *mbf1Δ* mutant. In fact, the level of frameshifted GFP/RFP protein per mRNA was very similar in the single *mbf1Δ* mutant to that in the double *asc1Δ mbf1Δ* mutant (*Figure 3A*, *Figure 3—figure supplement 1D*, *Supplementary file 2*). These results are most consistent with a single pathway of reading frame maintenance, which Asc1 influences.

We confirmed that the +1 GFP signal detected in our mutants was due to frameshifting rather than another aberrant translation event by directly measuring both the size and amount of GFP fusion protein. The amount of full-length GFP protein in the Western blot corresponds to the GFP/RFP values obtained from flow analysis (*Figure 3B*) indicating that +1 GFP/RFP signal in our mutants is due to frameshifting.

If the defect in the *asc1Δ* mutant that results in frameshifting is due to a failure of the Mbf1 pathway, then overproduction of Mbf1 in the *asc1Δ* mutant might suppress frameshifting in this mutant. We found that expression of *MBF1* on a multi-copy plasmid did suppress frameshifting in the *asc1Δ* strain to 1/3 that seen with an empty vector, but did not affect in-frame expression (*Figure 3C*). The overproduction of Mbf1 was not complementing a reduced abundance of Mbf1 in this mutant. We did not detect a reduction in Mbf1-HA (which complements the *mbf1Δ* mutant) in the *asc1Δ* strain (*Figure 3—figure supplement 2A*), although *asc1* mutants generally exhibit a defect in expression

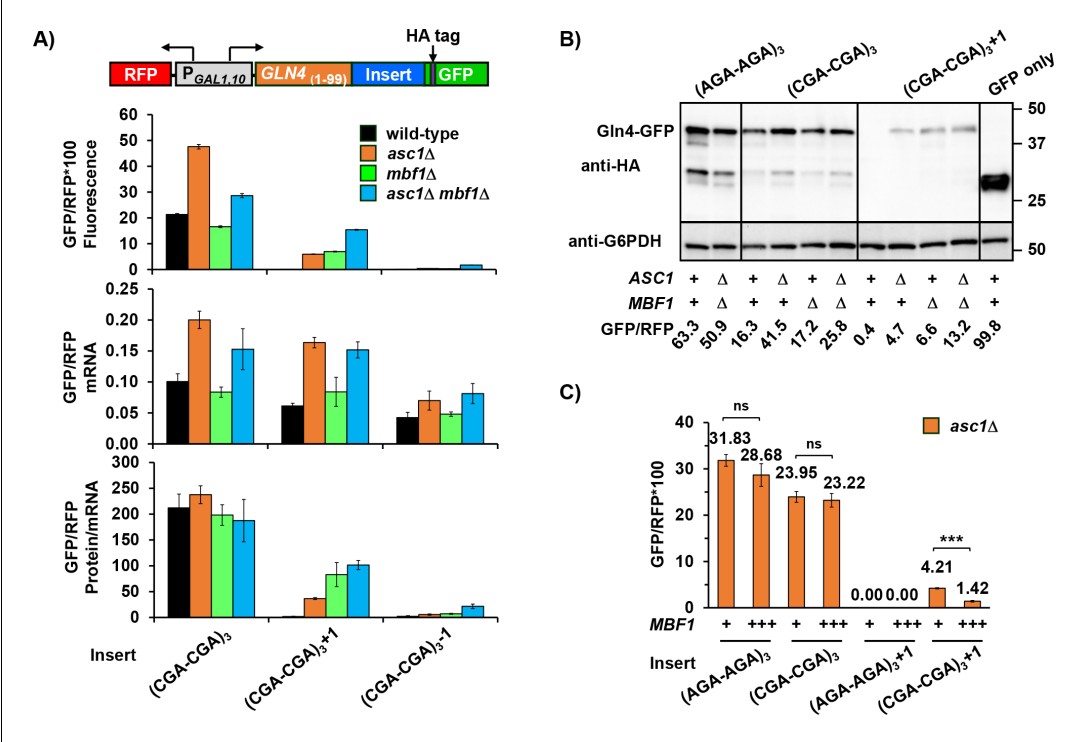

**Figure 3.** Mbf1 and Asc1 play distinct, but related, roles at CGA codon pairs. (A) Analysis of effects of *asc1Δ, mbf1Δ and asc1Δ mbf1Δ* mutations on protein expression (fluorescence), mRNA levels and protein/mRNA of *GLN4*(1-99)-GFP reporters containing three CGA-CGA codon pairs in the 0, +1, and −1 reading frames. GFP/RFP values are reported in all cases. (B) Western analysis of Gln4(1-99)-GFP fusion protein in yeast strains from (A) indicates the expression of frameshifted Gln4(1-99)-GFP full-length protein in all three mutants. The protein was detected by anti-HA antibody recognizing the HA epitope between the codon insert and GFP. The GFP and RFP values were measured by flow cytometry while harvesting for cell lysis. (C) Overproduction of Mbf1 suppressed frameshifting at CGA-CGA codon pairs in the *asc1Δ* mutant, but did not affect the in-frame expression, based on GFP/RFP expression from the indicated reporters shown in (A). ns: p > 0.05, ***p < 0.001.

DOI: https://doi.org/10.7554/eLife.39637.010

The following source data and figure supplements are available for figure 3:

**Source data 1.** Source data file for *Figure 3A and C*).

DOI: https://doi.org/10.7554/eLife.39637.013

**Figure supplement 1.** Analysis of effects of *asc1Δ, mbf1Δ and asc1Δ mbf1Δ* mutations on protein and mRNA expression of GFP reporters.

DOI: https://doi.org/10.7554/eLife.39637.011

**Figure supplement 1—source data 1.** Source data file for *Figure 3—figure supplement 1B*.

DOI: https://doi.org/10.7554/eLife.39637.014

**Figure supplement 2.** Frameshifting is likely not due to reduction in Mbf1 protein in the *asc1Δ* mutant nor to limiting Asc1 protein in the *mbf1Δ* mutant.

DOI: https://doi.org/10.7554/eLife.39637.012

**Figure supplement 2—source data 2.** Source data file for *Figure 3—figure supplement 2B*.

DOI: https://doi.org/10.7554/eLife.39637.015

of small proteins (*Thompson et al., 2016*). We also considered that *mbf1* mutants might require additional Asc1 protein, but additional copies of *ASC1* did not suppress frameshifting in an *mbf1Δ* mutant (*Figure 3—figure supplement 2B*). Thus, the frameshifted GFP/RFP fluorescence in the *asc1Δ* strain is likely a result of both an increase in mRNA and a defect in the Mbf1 pathway. We infer that Mbf1 and Asc1 contribute in distinct ways to the response to CGA codon pairs, but we do not know if Asc1 also has a direct role in the reading frame maintenance pathway.

## Mbf1 and Asc1 work at a common subset of inhibitory codon pairs and at a single inhibitory codon pair in a context-dependent manner

To address the mechanism of frameshifting and to understand the relationship between Asc1 and Mbf1, we set out to identify the protein and sequence requirements for efficient frameshifting. We began by examining frameshifting at 12 of 17 inhibitory codon pairs all of which cause reduced expression and many of which exhibit high ribosome occupancy, indicative of slow translation (*Gamble et al., 2016*), a common feature of many frameshifting sites.

We found that Mbf1 and Asc1 act on the same inhibitory codon pairs. Frameshifting occurs with high efficiency at three codon pairs (CGA-CGA, CGA-CGG, and CGA-CCG) in the *mbf1Δ* mutant (*Figure 4A*), the only three pairs at which Asc1 substantially modulates in–frame expression levels relative to synonymous optimal reporters (*Figure 4B*). As might be expected, frameshifted GFP/RFP for CGA-CCG and CGA-CGA is greater in the *asc1Δ mbf1Δ* mutant than in the *mbf1Δ* single mutant (*Figure 4A*, *Figure 4—figure supplement 1A*, *Supplementary file 2*). Surprisingly, this is not true for the CGA-CGG construct at which frameshifting is remarkably high (~76% based on data from *Figure 4—figure supplement 2C*). We address the source of this high frameshifting below and the lack of synergy in the Discussion.

Lower levels of frameshifting were also detected at 4 additional pairs (CGA-AUA, CGA-CUG, CGA-GCG, and CUC-CCG) in the *asc1Δ mbf1Δ* mutant, but not in the *mbf1Δ* single mutant (*Figure 4A*, *Figure 4—figure supplement 1B*). Analysis of these seven codon pairs indicates that frameshifting was detected at the seven most slowly translated codon pairs in the yeast genome [based on analysis in (*Gamble et al., 2016*)] and at every inhibitory pair with CGA in the 5' position, consistent with slow decoding of CGA in the P site (*Tunney et al., 2018*).

To test the idea that Asc1 and Mbf1 prevent frameshifting at any slowly translated sequence, we measured frameshifting at a sequence which forms a secondary structure to slow down translation and induces no-go mRNA decay (*Doma and Parker, 2006*; *Harigaya and Parker, 2010*; *Passos et al., 2009*). In accordance with this idea, frameshifted GFP/RFP protein from both +1 and −1 constructs was detectable in wild type, greater in each single mutant and even greater in the *asc1Δ mbf1Δ* mutant (*Figure 4C*). By contrast, these mutants did not affect frameshifting efficiency at the programmed frameshift site for *TY1* (*Figure 4—figure supplement 1C*) (*Belcourt and Farabaugh, 1990*). Thus, Mbf1 and Asc1 regulate reading frame maintenance at a translational pause (no-go site), but do not enhance frameshifting at site in which translational slippage is encoded.

To define the source of efficient frameshifting in the CGA-CGG reporter, which has 3 CGA-CGG pairs (*Figure 4A*), we initially determined that the first CGA-CGG codon pair was responsible for highly efficient frameshifting (*Figure 4—figure supplement 2A and B*). To define the sequence requirements for efficient frameshifting, we varied the sequences surrounding this single CGA-CGG pair and measured frameshifted GFP/RFP in the various mutants. Either of two changes to the sequence downstream of the CGA-CGG pair (one a point mutation and another a codon insertion) eliminated efficient frameshifting in all three mutant strains (*Figure 4D*). Furthermore, altering the two nucleotides downstream of the first codon pair in the three codon pair reporter reduced frameshifted GFP/RFP in all three mutants and also restored the synergistic dependence on *MBF1* and *ASC1* (*Figure 4—figure supplement 2C and D*). By contrast, none of three upstream changes (to the single CGA-CGG reporter) substantially reduced frameshifting (*Figure 4D*). Thus, the CGA-CGG-C 7-mer is required for efficient frameshifting.

To find out if frameshifting can occur at single CGA-CGG pairs in other sequence contexts, we tested sequences from seven yeast genes in our reporters, including six codons on either side of CGA-CGG pair. Frameshifted GFP/RFP was detected in *asc1Δ mbf1Δ* mutants in all cases (*Figure 4E*, *Figure 4—figure supplement 2E*). In particular, the two native sequences with CGA-CGG-C resulted in frameshifted GFP/RFP in the single mutants (*Figure 4E*, *Figure 4—figure supplement 2E*). Thus, CGA-CGG-C is likely a frameshifting sequence, and contexts that allow frameshifting have not been eliminated from native genes.

## +1 frameshifting occurs with the CGA codon in the P site

To understand how frameshifting occurs, we wanted to define the direction and position of the actual frameshift. The high efficiency of frameshifting at the CGA-CGG-CAC sequence provided a useful tool to study frameshifting since there is only a short potential frameshifting sequence (a

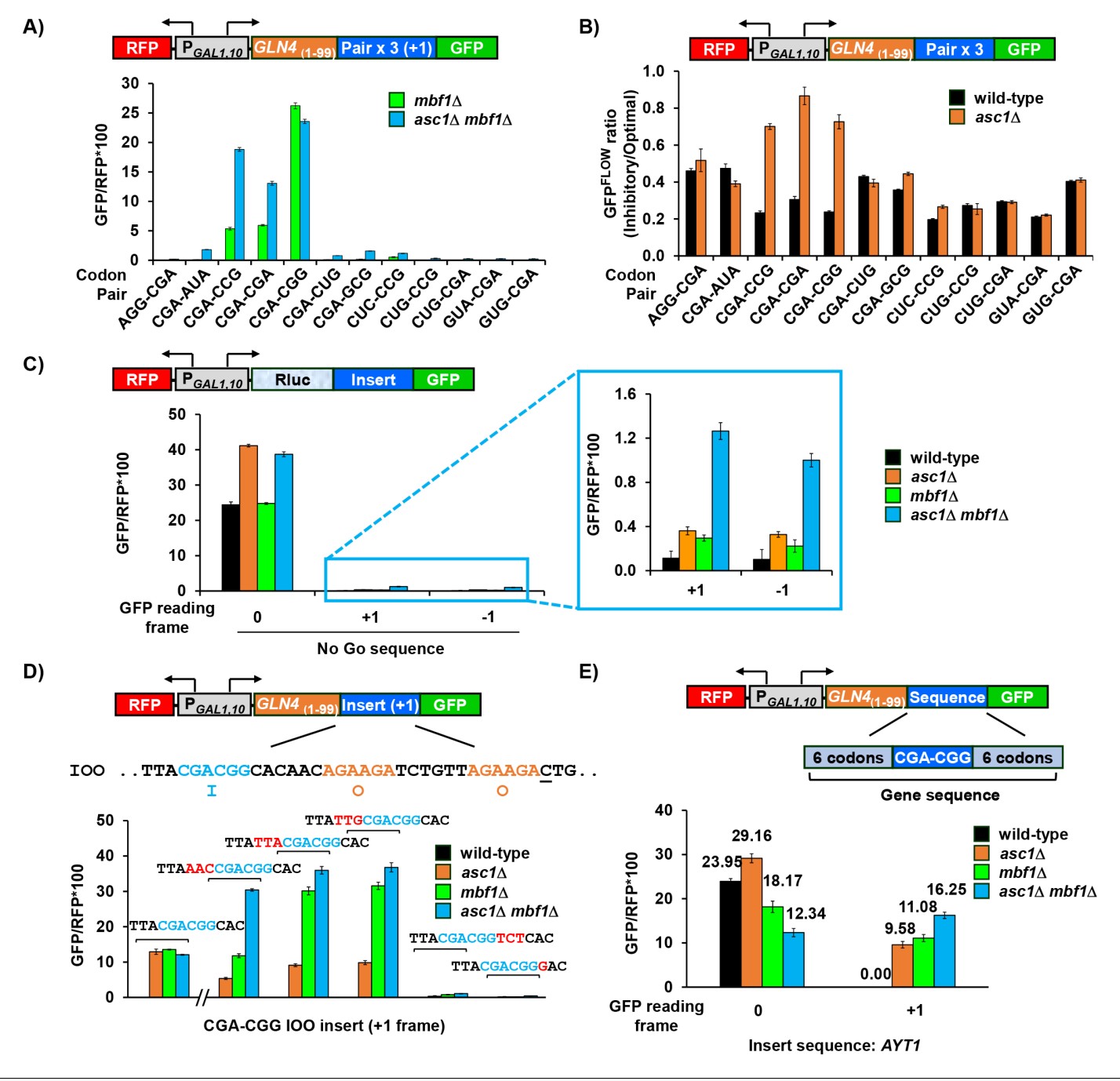

**Figure 4.** Mbf1 and Asc1 work at a common subset of inhibitory codon pairs and at a single inhibitory codon pair in a context-dependent manner. (**A**) Frameshifted GFP/RFP fluorescence was detected at three inhibitory codon pairs (*Gamble et al., 2016*) in the *mbf1*Δ mutant, and at seven codon pairs in the *asc1*Δ *mbf1*Δ double mutant. Frameshifting was assayed from reporters bearing 3 copies of the indicated inhibitory codon pair and a +1 nucleotide to place GFP in the +1 frame. (**B**) In-frame expression downstream of three inhibitory codon pairs (CGA-CGA; CGA-CCG; CGA-CGG) was improved by the deletion of *ASC1*. The GFP$^{FLOW}$ ratio is the ratio of GFP/RFP from reporters with three copies of an inhibitory pair relative to GFP/RFP from synonymous reporters with three copies of the optimized pair. (**C**) Mutation of either *ASC1* or *MBF1* allowed frameshifting at no-go sequences in the GFP reporter, and mutation of both *ASC1* and *MBF1* resulted in significantly more frameshifted GFP/RFP. (**D**) Variation of the sequences surrounding a single CGA-CGG inhibitory codon pair indicated that the 3' nucleotide downstream of the pair was required for efficient frameshifting. Frameshifted GFP/RFP from *GLN4*(1-99)-insert-+1 GFP reporters with a single CGA-CGG inhibitory codon pair was analyzed in the four indicated strains. Variations in the sequences surrounding this pair are shown in red. The construct, IOO contains the inhibitory CGA-CGG pair in position 1(I) and synonymous optimal pairs (AGA-AGA) in the positions 2 and 3 (OO) sites. The original IOO construct was measured separately (hash marks) and also reported in *Figure 4—figure supplement 2B*. (**E**) Analysis of effects of *asc1*Δ, *mbf1*Δ and *asc1*Δ *mbf1*Δ mutations on expression of *GLN4*(1-99)-GFP

*Figure 4 continued on next page*

*Figure 4 continued*

reporters containing the native yeast *AYT1* sequence with a single CGA-CGG codon pair in 0 and +1 reading frames. This native yeast sequence provoked significant amount of frameshifting.

DOI: https://doi.org/10.7554/eLife.39637.016

The following source data and figure supplements are available for figure 4:

**Source data 1.** Source data file for *Figure 4A–E*.

DOI: https://doi.org/10.7554/eLife.39637.019

**Figure supplement 1.** Analysis of effects of *asc1Δ, mbf1Δ and asc1Δ mbf1Δ* mutations on expression of different GFP reporters containing two sets of inhibitory codon pairs and programmed frameshifting site in *TY1*.

DOI: https://doi.org/10.7554/eLife.39637.017

**Figure supplement 1—source data 1.** Source data file for *Figure 4—figure supplement 1A-C*.

DOI: https://doi.org/10.7554/eLife.39637.020

**Figure supplement 2.** Efficient frameshifting occurs at a single CGA-CGG pair in a particular context.

DOI: https://doi.org/10.7554/eLife.39637.018

**Figure supplement 2—source data 2.** Source data file for *Figure 4—figure supplement 2B-E*.

DOI: https://doi.org/10.7554/eLife.39637.021

single inhibitory codon pair). We inserted this sequence with its neighboring codons from the RNA-ID reporter into a construct for purification of the frameshifted polypeptide (*Figure 5A*). The construct was designed such that the protein could be purified either with an upstream affinity tag (GST) to yield all polypeptides or with a downstream affinity tag (Strep II or the ZZ domain of IgG) to yield only frameshifted polypeptides. Treatment with LysC, which cleaves after lysine was expected to yield a 16 or 17 amino acid peptide for analysis by mass spectrometry, depending upon the mechanism of frameshifting.

If frameshifting occurred in the local region near the CGA-CGG codon pair, there are four possible events that could all give rise to +1 GFP signal. Ribosomes could frameshift in the +1 direction with either the CGA or the CGG in the P site, yielding the RGTT or the RRTT sequences shown in *Figure 5B*. Alternatively, ribosomes could undergo −2 frameshifting at either codon, yielding the peptides RDGTT or RRGTT (*Figure 5B*). In yeast, −2 frameshifting was observed upon expression of the mammalian antizyme (*Matsufuji et al., 1996*) and −2 frameshifting also occurs in PRRSV virus (*Fang et al., 2012*). We purified the frameshifted protein, as well as an in-frame control protein with the sequence expected for a −2 frameshift at CGG (*Figure 5C*) and subjected them to mass spectrometry. The frameshifted protein yielded the peptide VTNLRGTTWSHPQFEK, the expected peptide from a +1 frameshift beginning with the CGA codon in the P site of the ribosome. Thus, we infer that frameshift occurs with CGA in the P site, yielding only one Arg amino acid on the nascent peptide, then switches to a glycine codon GGC.

To determine if aminoacyl tRNA amounts affect frameshifting, we compared the effects of additional copies of specific Arg and Gly tRNAs on frameshifting in the *asc1Δ mbf1Δ* double mutant. We found that introduction of additional copies of the gene encoding tRNA$^{Arg(CCG)}$, which decoded the in-frame CGG codon, severely reduced frameshifting (*Figure 5D*), as expected if arg-tRNA$^{Arg(CCG)}$ competes with gly-tRNA$^{Gly(GCC)}$ for the A site. Similarly, we found that addition of extra copies of tRNA$^{Gly(GCC)}$ which decodes +1 frame GGC codon significantly increased frameshifting in our original CGA-CGG-CAC context, as might be expect if the GGC codon is used (*Figure 5D*). Additional copies of tRNA$^{Arg(ICG)}$, tRNA$^{Asp(GUC)}$, tRNA$^{His(GUG)}$, tRNA$^{Ser(AGA)}$ had little or no effect, as expected since none of the codons decoded by these tRNAs should be occupying the A site during frameshifting. These results indicate that the frameshifting occurs within the single CGA-CGG-CAC sequence and is modulated by the concentration of aminoacyl tRNAs decoding the out-of-frame codon.

## Discussion

We have uncovered a eukaryotic specific system that maintains the reading frame when ribosomes stall. Reading frame maintenance of stalled ribosomes is achieved in two ways: by direct inhibition of frameshifting; and by aborted translation coupled with mRNA decay. The system is composed of two proteins that lack bacterial homologs, the archaeal/eukaryotic Mbf1 protein and the eukaryotic ribosomal protein Asc1/RACK1, as well as one universally conserved ribosomal protein Rps3.

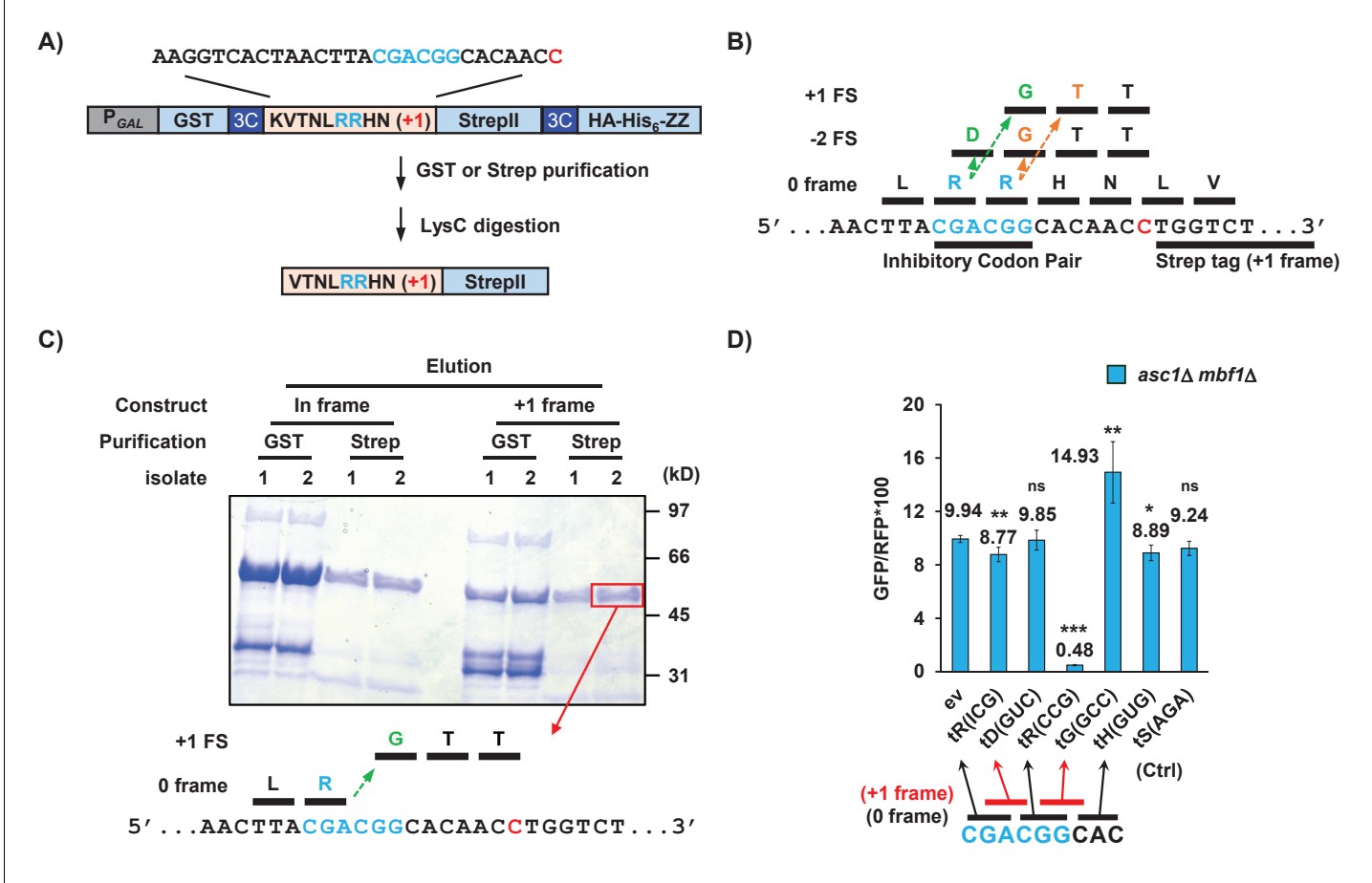

**Figure 5.** Frameshifting occurs in the +1 direction with the CGA codon in the P site and is modulated by tRNA competition at the A site. (**A**) Schematic of the purification construct for the frameshifted peptide. An eight amino acid sequence with a single CGA-CGG pair from the RNA-ID reporter was inserted between a GST tag and an out-of-frame StrepII tag. LysC treatment of purified frameshifted protein yields a 16 or 17 amino acid peptide. The red nucleotide indicates the extra nucleotide in the +1 frame construct. (**B**) Schematic of four possible frameshifting events at the inhibitory CGA-CGG codon pair, each of which can be distinguished by one or two amino acids in the resulting peptide. Ribosomes can frameshift either in the forward direction (+1) or in the reverse direction (−2) when the P site is occupied by either the CGA codon (first amino acid in the out-of-frame peptide shown in green) or the CGG codon (the first amino acid in the out-of-frame peptide shown in orange). (**C**) Purified protein products of both in-frame and +1 frame constructs were analyzed by SDS-PAGE, stained with Coomassie Blue. The frameshifted protein of +1 frame construct from Strep purification (in red box) was excised, cleaved with LysC and analyzed by Mass Spectrometry, resulting in identification of the peptide shown below the figure. This peptide corresponds to that expected of a +1 frameshift occurring when the CGA codon occupies the P site. (**D**) Overexpression of tRNA corresponding to +1 frame codon improved frameshifting efficiency, while overexpression of tRNA corresponding to the next in-frame codon significantly reduced frameshifting. ns: p > 0.05, *p < 0.05, **p < 0.01, ***p < 0.001.

DOI: https://doi.org/10.7554/eLife.39637.022

The following source data is available for figure 5:

**Source data 1.** (Source data file for 5D).

DOI: https://doi.org/10.7554/eLife.39637.023

Mutations in any of these proteins result in increased frameshifting at CGA codon repeats. More-over, the Rps3 residues in which mutations affect reading frame maintenance are specifically conserved in eukaryotes (and differ in archaeal and bacterial Rps3), consistent with a eukaryotic-specific mechanism. We suggest that when ribosomes stall (*Simms et al., 2017b*), two distinct sets of events occur: Asc1 triggers a set of responses that result in aborted translation, recruitment of the RQC complex and mRNA decay (*Brandman et al., 2012*; *Defenouillère et al., 2013*; *Shao et al., 2013*; *Simms et al., 2017b*; *Sitron et al., 2017*; *Verma et al., 2013*); while Mbf1 and Rps3 cooperate at the stalled ribosomes to prevent frameshifting (*Figure 6A*). In the absence of Mbf1 and Asc1, ribosomes frameshift efficiently, even at a single CGA-CGG pair in some cases, including sequences

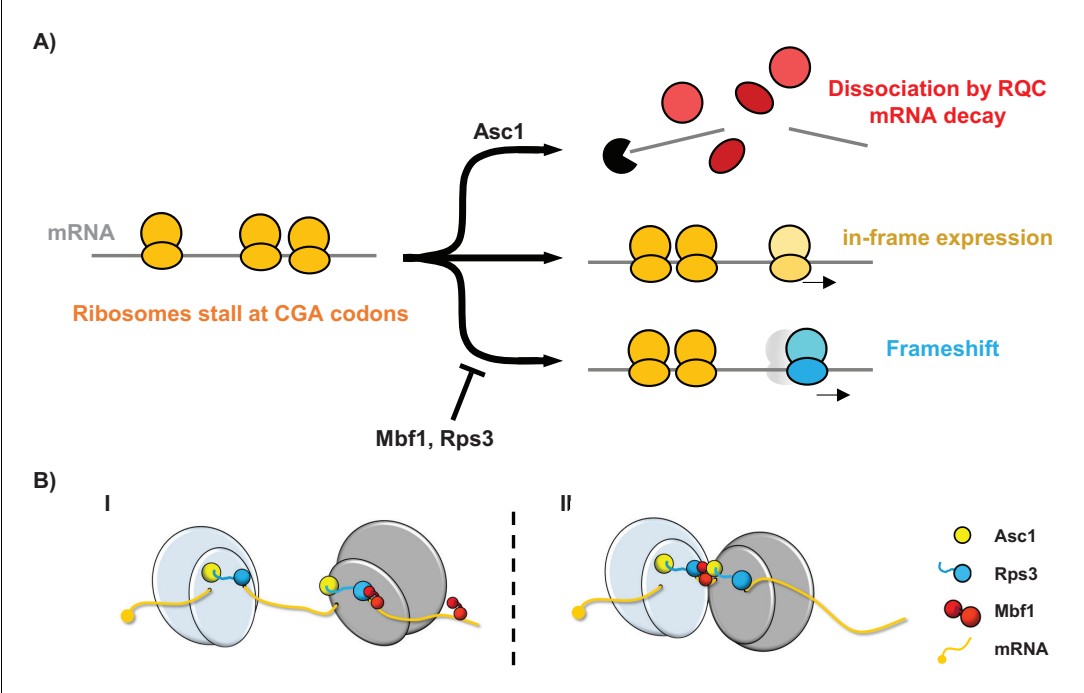

**Figure 6.** Models for the interplay between the Mbf1/Rps3 and Asc1-mediated RQC pathways and for the role of Mbf1 in reading frame maintenance. (**A**) Model of the two pathways that maintain reading frames of stalled ribosomes at CGA codons. Mbf1 and Rps3 act on stalled ribosomes to prevent frameshifting while Asc1 causes removal of many of these ribosome from the translating pool and destroys the mRNA with the stall sequence. (**B**) Two models of roles of Mbf1 and Rps3 in reading frame maintenance. In model I, Mbf1 (two domains shown in red) interacts with mRNA, and is recruited through an interaction with Rps3 (blue) to the leading stalled ribosome (gray) to restrict mRNA movement. The interaction must be transient, with removal of Mbf1 when the ribosome translocates. In model II, Mbf1 is recruited to the colliding ribosomes (light blue) possibly by both Asc1 (yellow) and Rps3. We postulate that again an interaction with mRNA could buffer the effects of ribosome collision.

DOI: https://doi.org/10.7554/eLife.39637.024

found in the native yeast genome. Frameshifting on the CGA-CGG codon pair occurs in the +1 direction, with the CGA codon in the P site of the ribosome and is modulated by availability of in-frame and +1 frame A site tRNAs.

The coordinated activities of both the Asc1 and Mbf1/Rps3 pathways are likely important to maintain the reading frame, since the absence of either pathway results in increased frameshifting. We think both pathways are likely engaged by similar stalls, since we noted evidence that Asc1 regulates expression (either in-frame or out-of-frame) at every sequence at which Mbf1 acted. However, there may be differences in the extent to which each pathway contributes at particular stall sites. For instance, at sequences at which frameshifting occurs rapidly (i.e. CGA-CGG-C), Mbf1 may play the critical role; in the absence of Mbf1, ribosomes frameshift before they can be captured by the Asc1 pathway. Generally, some fraction of ribosomes are removed from the translating pool with concomitant mRNA decay, while ribosomes that remain stalled are kept in-frame by Mbf1/Rps3 (*Figure 6A*).

The Asc1-mediated events couple mRNA decay to abortive translation (*Sitron et al., 2017*), thus effectively reducing the number and duration of ribosome stalls. The integration of RNA decay with translation is extensive: cleavage of mRNA occurs upstream of many ribosome stalls including at CGA repeats (*Chen et al., 2010*; *Doma and Parker, 2006*; *Guydosh and Green, 2014*; *Letzring et al., 2010*; *Simms et al., 2017b*) and is modulated by Asc1 (*Ikeuchi and Inada, 2016*; *Kuroha et al., 2010*). mRNA decay is triggered by many problems in translation, such as nonsense mediated decay (NMD), no-go and NonStop decay [See (*Shoemaker and Green, 2012*; *Simms et al., 2017a*) and by slow translation (*Presnyak et al., 2015*; *Radhakrishnan et al., 2016*). For reading frame maintenance, it seems likely that both reducing the number of stalled ribosomes (by aborting translation) and removing the mRNA are important. Deletion of *ASC1* in a *mbf1Δ* mutant results in an increase in frameshifted GFP/RFP protein that is directly proportional to the

increase in mRNA. However, *asc1Δ* mutants exhibit a 50-fold increase in frameshifting relative to wild type cells with less than a 3-fold increase in mRNA levels. One likely explanation for frameshifting in the *asc1Δ* mutant is that Mbf1 becomes limiting due to an increase in the number of stalled ribosomes; this idea is strengthened by the observation that overproduction of Mbf1 suppressed frameshifting in the *asc1Δ* mutant. Alternatively, Asc1 might also play a direct role in the reading frame maintenance function. Asc1 interacts with the C terminal region of Rps3 and could affect the conformation of Rps3 or its interaction with Mbf1. Mbf1 was found in the vicinity of Asc1 (*Opitz et al., 2017*) lending credence to the idea that Asc1 could interact with Mbf1 (although such an interaction cannot be obligatory).

We think Rps3 and Mbf1 inhibit frameshifting in a coordinated manner, perhaps due to their interactions with mRNA or to Mbf1's interaction with the ribosome. The role of Rps3 in this process is likely to involve interactions with either the incoming mRNA or proteins external to the ribosome. The *RPS3* mutations that affect frameshifting map to residues (*S104, L113, G121, K108*) on two α-helices or their connecting loop right next to the entering mRNA. Although this section of Rps3 is involved in helicase activity and initiation selectivity (*Dong et al., 2017*; *Takyar et al., 2005*), the residues mutated in frameshifting selections were not specifically those involved in these activities. Instead, these residues all sit on the solvent side of the ribosome and could form an interface interacting with mRNA or mRNA-bound proteins. The role of Mbf1 is likely mediated by interactions with either or both of the mRNA (*Beckmann et al., 2015*; *Klass et al., 2013*) and the ribosome (*Blombach et al., 2014*; *Opitz et al., 2017*). The apparent RNA binding domain maps to the less conserved N terminal domain (*Klass et al., 2013*), while ribosome binding activity of the archaeal homolog of Mbf1 maps to its C-terminal HTH domain and the linker at the N terminus of this domain, which are both conserved with eukaryotes (*Blombach et al., 2014*); our frameshifting mutations cluster in the conserved linker region of Mbf1. Moreover, Mbf1 is sufficiently abundant with ~85,000 molecules per cell to participate in general translation cycles, although it is less abundant than core ribosomal proteins (~200,000) (*Kulak et al., 2014*).

There are two reasonable models to account for the role of Mbf1 and Rps3 in reading frame maintenance (*Figure 6B*). The first model is that Mbf1 has a loose association with mRNA and is recruited to the leading stalled ribosome by an interaction with Rps3; the interactions with the ribosome and the mRNA at the stall site could restrict mRNA movement in the ribosome. Based on structures of prokaryotic ribosomes caught in translocation, mRNA flexibility may occur in ribosomes lacking an A site tRNA due to few contacts with the region of mRNA near the A site (*Zhou et al., 2013*), or due to a failure of two rRNA pawls that lock the mRNA in a translocating ribosome (*Zhou et al., 2013*), or due to defects in the interactions with elongation factor 2 (*Zhou et al., 2014*). In the absence of Mbf1, the ribosome stall might allow sufficient time for mRNA flexibility, resulting in frameshifting. The second model, which is based on the observation that ribosome collisions trigger no-go decay (*Simms et al., 2017b*), is that Mbf1 is recruited to colliding ribosomes to buffer the collision effects; in this case Asc1 and Rps3 might both participate in Mbf1 recruitment. Mbf1 could prevent ribosome collision-mediated movement of the leading ribosome on the mRNA.

Frameshifting occurs by a mechanism that involves the interplay between the two adjacent codons, in which I•A wobble interaction in the P site in conjunction with competition between tRNAs entering the A site results in the frameshift, consistent with a model proposed by Baranov *et al.* (*Baranov et al., 2004*). Two lines of evidence support this mechanism. First, we demonstrated that, in the *asc1Δ mbf1Δ* double mutant, ribosomes frameshift at a single CGA-CGG codon pair (in a particular context) when the CGA codon occupies the P site. We infer that CGA codon in the P site is generally important for frameshifting, because six of the seven codon pairs on which ribosomes frameshift are CGA-NNN and the three efficient pairs are CGA-CNN. The wobble interaction between the CGA codon and tRNA could weaken the interaction between mRNA and the ribosome, which in turn could slow down the elongation cycle. Second, we found that frameshifting is influenced by the abundance of the in-frame and out-of-frame tRNAs for next position, which implies that the frameshift occurs after translocation of the CGA from the A site to the P site. We speculate that the flexibility of the wobble base pair interaction between inosine and other nucleotides could actively facilitate the acceptance of out-of-frame A site tRNA. For instance, we consider that a rare instance in which the A base in CGA is bulged out might be stabilized by the very strong I•C interaction, increasing the time available to accept the out-of-frame tRNA.

The eukaryotic specific reading frame maintenance activity, involving Mbf1 and ribosomal proteins Rps3 and Asc1, is likely to be important for translation accuracy in the yeast genome. Mutations in either *RPS3* or *MBF1* suppressed frameshifting mutations in several native yeast genes (*Hendrick et al., 2001*). Moreover, mutations in *MBF1* and *ASC1* resulted in detectable frameshifting in a set of native gene sequences with only a single inhibitory codon pair flanked by six adjacent codons on each side, although it is apparent that the frameshifting potential within a particular sequence is not simply due to the presence of a single inhibitory codon pair. These results confirmed that Mbf1 with Rps3 and Asc1 play a critical role in maintaining the reading frame during normal translation cycles. It is still unknown why this eukaryote-specific reading frame maintenance system evolved and why it is important to eukaryotes, but not bacteria.

# Materials and methods

## Key resources table

| Reagent type | Designation | Source or reference | Identifiers | Additional information |
|---|---|---|---|---|
| Gene (*Saccharomyces cerevisiae*) | *MBF1/ YOR298C-A* | | SGD:S00 0007253 | |
| Gene (*S. cerevisiae*) | *ASC1/ YMR116C* | | SGD:S00 0004722 | |
| Gene (*S. cerevisiae*) | *RPS3/ YNL178W* | | SGD:S00 0005122 | |
| Strain, strain background (*S. cerevisiae, MATa*) | BY4741 | Open Biosystems | | |
| Strain, strain background (*S. cerevisiae, MATα*) | BY4742 | Open Biosystems | | |
| Genetic reagent (*S. cerevisiae*) | See Supplementary table 1 in *Supplementary file 3* | this paper | | |
| Antibody | anti-HA High Affinity (Rat monoclonal) | Roche | 3F10, catalog# 11867423001; RRID:AB_10094468 | 1:3000 dilution |
| Antibody | Goat anti-Rat IgG-HRP conjugate | Jackson Immuno Research | catalog# 112-035-003; RRID:AB_2338128 | 1:5000 dilution |
| Antibody | anti-G-6-PDH (Rabbit monoclonal) | Sigma | catalog# A9521; RRID: AB_258454 | 1:5000 dilution |
| Antibody | Goat anti-Rabbit IgG-HRP conjugate | Biorad | catalog# 1706515; RRID:AB_11125142 | 1:10000 dilution |
| Recombinant DNA reagent | See Supplementary table 2 in *Supplementary file 3* | this paper | | |
| Sequence-based reagent | See Supplementary table 3 in *Supplementary file 3* | this paper | | |
| Sequence-based reagent | Random Primers | Invitrogen | catalog# 48190011 | |

*Continued on next page*

*Continued*

| Reagent type | Designation | Source or reference | Identifiers | Additional information |
|---|---|---|---|---|
| Commercial assay or kit | T4 DNA Polymerase, LIC-qualified | Novagen | catalog# 70099 | Used for ligation-independent cloning |
| Commercial assay or kit | Super Script II Reverse Transcriptase | Invitrogen | catalog# 18064014 | |
| Commercial assay or kit | RQ1 Rnase-Free Dnase | Promega | catalog# M6101 | |
| Commercial assay or kit | RiboMAX Large Scale RNA Production System-T7 | Promega | catalog# P1300 | |
| Commercial assay or kit | MicroSpin G-25 columns | GE | catalog# 27-5325-01 | |
| Commercial assay or kit | Fast SYBR Green Master Mix | Applied Biosystems | catalog# 4385612 | |
| Commercial assay or kit | Glutathione sepharose-4B | GE | catalog# 17-0756-01 | |
| Commercial assay or kit | MagStrep 'type3' XT beads | IBA | catalog# 2-4090-002 | |
| Chemical compound, drug | 5-FOA | USBiological | catalog# F5050 | |
| Chemical compound, drug | Complete mini EDTA-free protease inhibitor | Roche | catalog# 118361 70001 | |
| Chemical compound, drug | leupeptin | Roche | catalog# 11017 128001 | |
| Chemical compound, drug | pepstatin | Roche | catalog# 11359 053001 | |
| Chemical compound, drug | Glutathione | Sigma | catalog# G4251 | |
| Chemical compound, drug | Biotin | IBA | catalog# 2-1016-002 | |
| Software, algorithm | MultAlin | (*Corpet, 1988*) | http:// multalin. toulouse. inra.fr/multalin/ | |

## Strains, plasmids, and oligonucleotides

Strains, plasmids, and oligonucleotides used in these studies are listed in *Supplementary file 3*, Tables 1-3. Parents for all yeast strains described in this study were either BY4741 (*MATa his3Δ1 leu2Δ0 met15Δ0 ura3Δ0*) or BY4742 (*MATαhis3Δ1 leu2Δ0 lys2Δ0 ura3Δ0*) (Open Biosystems). The $GLN4_{(1-99)}$-$(CGA)_6$+1-*URA3* reporter used in the selection was constructed with PCR-amplified DNAs (using oligonucleotides OJYW085, 086, 041, 089, 095 and 099), assembled by Ligation Independent Cloning (LIC) methods (*Alexandrov et al., 2004*; *Aslanidis and de Jong, 1990*) and then integrated into the *CAN1/YEL063C* locus on the chromosome V, selecting for canavanine-resistance; constructs

were checked by sequencing of genomic PCR fragments. RNA-ID reporters were constructed as described previously and integrated at the *ADE2* locus, using selection with *MET15* marker in *MATa* strains or *S.pombe HIS5* marker in *MATα* strains (*Dean and Grayhack, 2012*; *Gamble et al., 2016*; *Wolf and Grayhack, 2015*).

Yeast strains bearing *MBF1* deletions were constructed by amplification of the *kan*[R] cassette in the yeast strain from the corresponding knockout strain in the systematic deletion collection (Open Biosystems) (*Giaever et al., 2002*). The *MATa* yeast strain bearing a deletion of *RPS3* was constructed by amplification of *ble*[R] cassette (*Gueldener et al., 2002*) (oligos OW443 and OW445) and integration of this DNA into a strain bearing an *URA3* [*RPS3*] covering plasmid (pEAW433). Yeast strains bearing deletions of *ASC1* marked with the *S. pombe HIS5* marker (AW768), which have been described previously (*Wolf and Grayhack, 2015*), were constructed and maintained in the presence of a plasmid born copy of *ASC1* on a 2μ, *URA3* plasmid. To obtain the *asc1Δ* strain from the selection parent strain, the *ASC1* gene was deleted by a *ble*[R] cassette obtained by PCR amplification with oligos OW125 and OW126.

Plasmids bearing the *MBF1* gene were constructed by amplification of chromosomal *MBF1* gene from −580 in 5' UTR to +300 in 3' UTR with oligos OJYW124 and OJYW125, followed by cloning into the 2μ, *LEU2* vector (pAVA0577) and into the *CEN*, *LEU2* vector (pAVA0581) to create pEJYW203 and pEJYW176 respectively. The chromosomal HA-tagged *MBF1* was constructed by PCR amplification of HA-*kan*[R] sequence from pYM45 (Euroscarf) (*Janke et al., 2004*) with oligos OJYW130 and OJYW132, bearing homology to *MBF1*, followed by integration into the *MBF1* locus. This *MBF1-HA Kan*[R] cassette from −580 in 5'UTR to +300 in 3'UTR of *MBF1* (+1992 including *Kan*[R] sequences) was amplified from the chromosome with oligos OJYW157 and OJYW158, cloned into the XmaI and NheI sites in Bluescript as pEJYW279. The *mbf1* point mutations K64E and I85T were individually introduced into the plasmid pEJYW279 to make pEJYW302 and pEJYW307 respectively. The *mbf1-K64E* cassette was directly PCR-amplified from the mutant strain YJYW290-P38 with oligos OJYW157 and OJYW158 followed by digestion with XmaI and BamHI and integration into these two sites on pEJYW279. The *mbf1-I85T* mutation was introduced by PCR amplification from *MBF1-HA* cassette with OJYW170, which contains the mutation, and OJYW166, followed by integration into pEJYW279 between BamHI and AatII sites. Reconstructed *mbf1* point mutants were introduced into YJYW2566 (BY4741, *HIS3*[+]) with XmaI/NheI digested pEJYW302 and pEJYW307 selecting with *Kan*[R] marker. The plasmid template for in vitro transcription of GFP and RFP fragments (pEJYW407 and pEJYW409) was constructed by PCR-amplifying pEAW315 with oligos OJYW295/OJYW296 (for GFP) or OJYW297/OJYW299 (for RFP) followed by digestion with SphI and XmaI and integration into these two sites on pSP73 (Promega, cat.# P2221). Plasmids expressing tRNAs were obtained from Phizicky and Grayhack lab stocks (*Guy et al., 2012*; *Han et al., 2015*; *Letzring et al., 2010*).

## Selection for frameshifting mutants and identification of mutations

Ura[+] mutants were selected from 40 independent cultures of each *MATa* and *MATα* parent strains (YJYW289, YJYW329), and then were analyzed by flow cytometry to measure GFP and RFP expression. Ura[+] GFP[+] mutants, indicative of increased frameshifting efficiency, were selected for further study, with an emphasis on mutants that exhibited higher levels of frameshifting, that is GFP/RFP >4, (28% *MATα* and 66% *MATa* mutants). Diploids between 12 *MATa* mutant and 20 *MATα* mutants were created by mating in YPD for 2 hr at 25°C and selection on SD-Lys-Leu-His media for diploid cells, followed by streaking for single colonies. Then overnights of the resultant diploids and their haploid parents were spotted on SD-Leu and SD-Leu-Ura plates, which were grown at 30°C.

To identify the relevant mutation in YJYW290-P25, we obtained the Leu[-] derivative of this mutant (YJYW315) by screening replica plated single colonies from an overnight in YPD on YPD and SD-Leu plates. The Ura[+]/FOA-sensitive phenotype of this mutant was complemented with a genomic tiled library (*Jones et al., 2008*), selecting for FOA-resistant cells. First, 17 pools of DNA, each of which contained 96 plasmids (*Jones et al., 2008*), were transformed individually with >1000 colonies per plate. Transformants of each pool were then scraped and saved in 2 ml YPD +8% DMSO. These saves were plated based on their OD$_{600}$ (2 × 10$^7$ cells/OD$_{600}$ x ml) to obtain approximately 5,000 cells on SD-Leu and 50,000 cells on SD-Leu +0.5 xFOA. For 16 of 17 pools, there were no colonies on the FOA plates, while transformants of pool 15 had 330 FOA-resistant colonies with 1404 colonies on SD–Leu plate, corresponding to FOA-resistance for 2.3% cells. The plasmids responsible for FOA-resistance was identified by complementing with plasmids from individual rows and columns in

this pool as described above, followed by complementation with individual plasmids. Two plasmids from this pool conferred FOA-resistance and share a single gene, *MBF1*. The *MBF1* gene in 19 recessive mutants was amplified from their genomic DNA with oligos OJYW124 and OJYW125, followed by sequencing to confirm the mutated residues.

Whole genome sequencing on two dominant *MATα* mutants was performed to identify the mutated genes. For each strain,~30 $OD_{600}$ yeast cells were harvested and re-suspended in 1 ml prep buffer (2% Triton X-100, 1% SDS, 100 mM NaCl, 10 mM Tris-Cl pH 8.0, 1 mM EDTA) with ~1.5 g Zirconia/Silica beads (from BioSpec, catalog# 11079105z) and 1 ml PCA pH 8.0. The suspension was then vortexed at top speed for 3 min and mixed with 1 ml TE pH 8.0, followed by centrifugation in prespun PLG tubes (from 5prime, catalog# 2302830). Nucleic acids in the aqueous layer were ethanol precipitated with 5 ml 100% ethanol, followed by freezing on dry ice and centrifugation for 20 min at 4,000 rpm at 4°C. The pellet was re-suspended in 200 µl TE and incubated at room temperature for 1 hr with 0.2 µg/µl RNaseA to remove RNA contamination, followed by addition of 200 µl 1 M Tris-Cl pH 8.0, 2 µl of 5 mg/ml glycogen and 400 µl PCA, and centrifugation for 2 min at top speed at 4°C. The aqueous layer (~360 µl) was precipitated with 720 µl 100% ethanol and frozen on dry ice for 15 min; resulting pellets were re-suspended in 100 µl TE pH 8.0 and 100 µl 1 M Tris-Cl pH 8.0, followed by precipitation again with 400 µl 100% ethanol. The DNA pellet was then washed with 500 µl 70% ethanol and finally re-suspended in 50 µl sterile $ddH_2O$. Whole genome sequencing was performed by the UR Genomics Research Center resulting in *RPS3* mutations in these two *MATα* mutants. Mutations in two *MATa* dominant mutants were then identified by amplification of *RPS3* cassette with oligos OJYW159 and OJYW210, followed by sequencing.

## Analysis of yeast growth

Appropriate control strains (previously studied) and 2–4 independent isolates of each strain being tested were grown overnight at 30°C in media indicated, diluted to obtain $OD_{600}$ of 0.5, then serially diluted 10-fold twice; 2 µl diluted cells were then spotted onto the indicated plates and grown at different temperatures for at least two days.

## Flow cytometry

To examine mutants in either *RPS3* or *ASC1*, reporters were introduced into sets of strains bearing an *URA3* covering plasmid with either *RPS3* or *ASC1*, depending upon the chromosomal deletion. All sets of strains in a given panel contained the same *URA3* plasmid. Prior to analysis of GFP expression, strains were streaked on FOA containing plates, then single colonies were grown for analysis by flow cytometry.

Yeast strains bearing the modified RNA-ID reporters were grown overnight at 30°C in YP media (for strains without plasmid) or appropriate synthetic drop-out media (for strains with plasmid) containing 2% raffinose + 2% galactose + 80 mg/L Ade. The cell culture was diluted in the morning such that to the culture had a final $OD_{600}$ between 0.8–1.0. Analytical flow cytometry and downstream analysis were performed for four independent isolates of each strain (Outliers were rejected using a Q test with >90% confidence level) as previously described (*Dean and Grayhack, 2012*). Each flow experiment was also performed with proper controls including a GFP[-], RFP[+] strain. The GFP/RFP value from this control strain was subtracted from all tested strains on the same day to show signals above background (negative values are set to 0). P values were calculated using a one-tailed or two-tailed homoscedastic t test in Excel, as indicated in the source data for relevant figures.

## Western blotting

Western analysis of the GFP fusion proteins in the modified RNA-ID reporter and Mbf1 protein in yeast strains were performed with anti-HA antibody as described previously (*Gelperin et al., 2005*).

## RT-qPCR

mRNA measurements with reverse transcription (RT) reaction and quantitative PCR were performed as described previously (*Gamble et al., 2016*) with one significant difference. Quantification of mRNA was performed using in vitro transcribed GFP and RFP mRNA fragments, synthesized from linearized plasmid pEJYW407 and pEJYW409 using RiboMAX Large Scale RNA Production System-T7 (Promega, cat.# P1300). The synthesis reaction was followed by DNase treatment to remove the

DNA template and by elution through MicroSpin G-25 columns (GE, cat.# 27-5325-01) to remove unincorporated nucleotides. The synthesized RNA sample was analyzed by ultraviolet light absorbance at 260 nm on a nanodrop to determine the concentration and by electrophoresis to assess integrity. Each qPCR plate contained 5-point 1:5 dilution standard curves for both GFP and RFP, which were optimized to ensure that all samples fall into the linear range of the curves. For each tested strain, three biological replicates were analyzed.

## Purification of frameshifted peptide

To purify the frameshifted peptide from yeast, a *LEU2* plasmid containing either in-frame or +1 frame protein purification constructs were transformed into the *asc1Δ mbf1Δ* strain (YJYW378). Two independent transformants (FOA treated) of each construct were grown overnight in SD-Leu media and transferred into 80 ml S-Leu + 2% raffinose media in the morning. After reaching an $OD_{600}$ of 0.8–1.2, expression of the GST-StrepII-ZZ construct was induced by addition of 40 ml 3xYP + 6% galactose and growth was continued for 10 hr. Cells were collected by centrifugation and cell pellets were quick frozen on dry ice. The cell pellets were re-suspended in 1 ml extraction buffer (50 mM Tris-Cl pH 7.5, 1 mM EDTA, 4 mM $MgCl_2$, 5 mM DTT, 10% Glycerol, 1 M NaCl, 2.5 µg/ml leupeptin, 2.5 µg/ml pepstatin) and lysed with bead beating (10 repeats of 20 s beating followed by 1 min on ice), essentially as described previously (Quartley et al., 2009). The cell lysate was collected from the bead beating tubes by puncturing the bottom with a hot needle and blowing with low pressure air. Solid contents were removed by centrifugation before the remaining lysate was divided into half and purified on either GSH or Streptactin resin.

For GST purification: the cell lysate was first diluted with equal volume No Salt Wash Buffer (50 mM Tris-Cl pH 7.5, 4 mM $MgCl_2$, 5 mM DTT, 10% Glycerol) to bring the salt to 0.5 M NaCl. GSH resin [Glutathione sepharose-4B from GE, catalog# 17-0756-01; pre-washed with Wash Buffer (No Salt Wash Buffer + 0.5 M NaCl)] (50 µl/ ml of lysate) was added to the diluted cell lysate and the mixture was nutated for 3 hr at 4°C. The resin was separated from the liquid by centrifugation at low speed (<3,000 rpm) and washed twice with 0.5 ml Wash Buffer followed by 20 min nutation. The bound protein products were then eluted by nutating for 40 min with 100 µl Elution Buffer (Wash buffer + 20 mM NaOH + 25 mM glutathione); the elution step was repeated to increase the yield.

For Strep purification: the cell lysate was diluted with 5x volumes No Salt Wash Buffer (100 mM Tris-Cl pH 7.5, 1 mM EDTA, 2.5 µg/ml leupeptin, 2.5 µg/ml pepstatin) to bring the salt to 150 mM NaCl. MagStrep 'type3' XT beads [from IBA, cat# 2-4090-002; pre-washed with Wash Buffer (No Salt Wash Buffer + 150 mM NaCl)] were added to the diluted cell lysate (80 µl/ 3.3 ml diluted cell lysate). After nutating for 2 hr at 4°C, resins were separated from liquid using a magnetic separator, then the resin was washed with 1 ml Wash Buffer three times without additional incubation. The bound protein products were then eluted by adding 50 µl Elution Buffer (Wash buffer + 50 mM biotin) and nutating for 10 min, followed by separation using magnetic separator; the elution step was repeated to increase the yield.

## Mass spectrometry

The elution samples from both GST and Strep purification were analyzed by SDS-PAGE, followed by staining with Coomassie Blue. The bands from Strep purification of both in-frame and +1 frame constructs were excised and analyzed on the Q Exactive Plus Mass Spectrometer in the Mass Spectrometry Resource Center of the University of Rochester Medical Center.

## Acknowledgements

We thank Eric Phizicky, Christina Brule, Andrew Wolf and Lu Han for discussions of the science and comments on the manuscript; Christina Brule and Blake Bentley for assistance with experiments, Wendy Gilbert and Mary Thompson for assistance with ribosome profiling data. This research has been facilitated by the services and resources provided by the University of Rochester Mass Spectrometry Resource Laboratory and NIH instrument grant (1S10OD021486-01). We thank the University of Rochester Genomics Research Center for performing high-throughput sequencing library construction, sequencing, and primary data analysis for this study. We also thank the URMC Flow Cytometry Resource for technical support. This work was supported by NIH grant R01 GM118386 to EJG.

## Additional information

### Funding

| Funder | Grant reference number | Author |
| --- | --- | --- |
| National Institutes of Health | R01 GM118386 | Elizabeth J Grayhack |

The funders had no role in study design, data collection and interpretation, or the decision to submit the work for publication.

### Author contributions

Jiyu Wang, Conceptualization, Data curation, Formal analysis, Investigation, Writing—original draft, Writing—review and editing; Jie Zhou, Investigation, Assisted in analysis of native yeast sequences; Qidi Yang, Investigation, Assisted in analysis of Mbf1-Rps3 interaction; Elizabeth J Grayhack, Conceptualization, Data curation, Formal analysis, Supervision, Funding acquisition, Methodology, Writing—original draft, Writing—review and editing

### Author ORCIDs

Jiyu Wang (iD) http://orcid.org/0000-0002-1283-2934
Elizabeth J Grayhack (iD) https://orcid.org/0000-0003-2400-5490

### Decision letter and Author response

Decision letter https://doi.org/10.7554/eLife.39637.030
Author response https://doi.org/10.7554/eLife.39637.031

## Additional files

### Supplementary files

• Supplementary file 1 Related to *Figure 2*. Effects of *RPS3-K108E*, *mbf1Δ* and *RPS3-K108E mbf1Δ* mutants on GFP/RFP protein, mRNA and protein/mRNA from in-frame and frameshifted *RLuc*-Arg$_4$-GFP/RFP.
DOI: https://doi.org/10.7554/eLife.39637.025

• Supplementary file 2 Related to *Figure 3*. Effects of *asc1Δ*, *mbf1Δ* and *asc1Δ mbf1Δ* mutations on GFP/RFP protein, mRNA and protein/mRNA from *GLN4*$_{(1-99)}$-GFP reporters with three CGA-CGA codon pairs in the 0, +1, and −1 reading frames.
DOI: https://doi.org/10.7554/eLife.39637.026

• Supplementary file 3 Related to Key Resources Table in Materials and methods Strains, plasmids and oligonucleotides used in these studies.
DOI: https://doi.org/10.7554/eLife.39637.027

• Transparent reporting form
DOI: https://doi.org/10.7554/eLife.39637.028

### Data availability

All data generated or analyzed during this study are included in the manuscript and supporting files.

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
