## [Decision Letter]

Thank you for submitting your article "Multi-protein Bridging Factor 1(Mbf1), Rps3 and Asc1 prevent stalled ribosomes from frameshifting" for consideration by *eLife*. Your article has been reviewed by four peer reviewers and the evaluation has been overseen by a Reviewing Editor and James Manley as the Senior Editor. The reviewers have opted to remain anonymous.

The reviewers have discussed the reviews with one another and the Reviewing Editor has drafted this decision to help you prepare a revised submission.

We have received the comments from four expert reviewers all of whom found the genetic screen and its follow-up to contain interesting new experiments focused on the activities of the ribosome in the face of elongation distress. These studies argue that a factor, Mb1, previously implicated in transcription and translation, and the ribosomal protein S3 play a critical role in preventing frameshifting at sites of stalling. In the absence of frameshifting, when severe stalling occurs, the ribosome-associated protein Asc1 triggers a diverse set of quality control events that have been studied in some substantial detail by the field. While all the reviewers felt that this study nicely leveraged genetics to reveal unanticipated players in the ribosome-centric event, all were somewhat concerned that the study did not go sufficiently beyond the reporter-based screen to reveal molecular insights into the roles of these factors on the ribosome. There was extensive use of reporters, with very similar disruptive features, to detail the subtleties in the response to pause sequences, for example. The figures detailing the di-codon pair experiments (Figure 4) and experiments focused on a natural context (Figure 5) provided little new information and potentially could be placed in a supplemental figure. The most critical set of experiments that brought all the ideas in the manuscript together was Figure 6, where depletion and over-expression of tRNAs and mass spec were used to determine that the frameshifting is triggered only when CGA is in the P site with its associated I:A wobble base pairing.

In addition to some concerns that molecular mechanism was lacking, the reviewers all had concerns about the relative lack of analysis and discussion of the role of mRNA decay in the events being studied. As you will read in the detailed comments, the reviewers felt it is critical to generate RNA quantitation data on both the GFP and RFP mRNAs, and then to present the data in terms of these measurements. It is clear that there are some substantial effects on mRNAs levels that result from the various genetic mutations, and it likely that these effects do not account for the observed increases in GFP only on the +1 constructs. To support the strength of this conclusion, the RNA levels should be quantitated both for the (CGA)4 constructs as well as the (CGA)4+1 constructs, as the RNA changes will likely be equivalent, and there will only be increases in GFP on the +1 constructs. While we are relatively confident that the conclusions will hold, given the extent to which translation and decay are coupled in these pathways, this analysis seems critical, and a careful discussion important for the reader.

In addition to this main criticism (related to measuring RNA levels carefully), there are some smaller points to be addressed in the reviewer comments. For example, the data and discussion on Hel2/Slh1 seemed over-interpreted or unclear. At a glance, there are no substantial differences that result when these factors are deleted. Moreover, the discussion of how these pathways might intersect or what these data might mean were not easy to follow. The manuscript would benefit from removal from the main text of figures that add little to our understanding of the role of these new factors (Mbf1 and S3) but simply support the results of the initial screen (yeast growth plates and the very detailed study in Figure 5, for example). What would benefit from more discussion is a more complete analysis about how these results fit in with the literature on mRNA decay, and most importantly, some thoughts about what Mbf1 might be doing. A model figure with the relative positions of S3 and Asc1 on the ribosome might help frame such a discussion, even in the absence of new biochemical information on Mbf1.

In light of the strength of the screen and the identification of an interesting new player in ribosome quality control, the reviewers support publication in *eLife* if the various concerns are all addressed.

While the study fails to provide substantive mechanistic insights, we feel that the screen and follow-up to confirm the role of Mbf1 in preventing frameshifting will be a valuable contribution to the field.

Thank you for submitting your excellent work to the journal.

*Reviewer #1:*

In this paper, Wang and colleagues study protein factors and pathways that regulate translational frameshifting of stalled ribosomes. Previous work by the Grayhack lab identified Asc1 as a regulator of translational frameshifts for ribosomes decoding translation-inhibitory codons. Using a forward genetic screen in budding yeast, Wang and co-authors discover two more players in this process: Rps3 and Mbf1. The authors use genetic manipulations, translational reporters, and mass spectrometry to identify the contributions of different pathways to translational frameshifting and gain some mechanistic insight into how this process occurs.

The identification of the Mbf1/Rsp3 system of translational frame maintenance is an important contribution to the field and opens up new areas of research. Furthermore this work represents a systematic genetic dissection of the determinants of translational frameshifting, and the authors confirm the conclusions informed by synthetic stalling sequences using endogenous genomically-encoded sequences. However, additional and critical controls are necessary to support the authors' conclusions. Specifically, this work relies heavily on a ratiometric, fluorescent reporter system based on a bidirectional pGAL1,10 promoter. In this reporter, two separate mRNAs are created, one to use as an expression control (RFP) and one as an experimental condition that encodes various test sequences upstream of a GFP in different frames. By comparing protein expression levels between the two messages, the authors hope to quantify the effect of the experimental conditions on translation. Yet this system also is sensitive to changes that affect mRNA levels of sequences that stall ribosomes. In fact, all the results reported in this study could be the result of how Asc1, Mbf1, and Rps3 affect decay of mRNAs that encode stalls and frameshifts. This is more than a hypothetical possibility, as *asc1∆* has been reported to affect mRNA levels of messages with stalls (Kuroha et al., 2010; Ikeuchi et al., 2016; Sitron et al., 2017).

The paper is fit for publication in *eLife* if the authors can address this major concern as detailed below.

1) The pGAL1,10-RFP/insert-GFP reporter system needs additional validation to ensure that measurements of GFP/RFP values truly report on increases in the frequency of stall readthrough or frameshifting rather than mRNA metabolism. This validation is of particular concern due to the data presented in Figure 3D. We plotted the mRNA levels of the (CGA-CGA)3+1 reporter against the corresponding GFP/RFP fluorescence values and found there is an almost perfect linear relationship between the levels of GFP mRNA and GFP/RFP fluorescence in the wt, *asc1∆, mbf1∆*, and *asc1∆mbf1∆* strains. The slope we plotted is not 1, however, since we do not have the standard curves for these primer sets, we cannot know the absolute differences in mRNA, only the relative increase. Thus, the increase in GFP/RFP fluorescence in the mutant strains could be driven primarily by an increase in mRNA levels of the stalling transcript rather than increased frameshifting frequency.

To validate the reporters used in the context of frameshifting and stalling, the authors must measure GFP and RFP mRNA levels of (CGA-CGA)3, (CGA-CGA)3+1, (CGA-CGA)3-1 reporters using absolute quantification (not relative quantification) in wt, *asc1∆, mbf1∆*, and *asc1∆mbf1∆* strains by RT-qPCR.

- Include a standard curve for each primer set (varying cDNA input) to map experimental conditions back to absolute input levels. Ensure the assay is conducted in the linear range of the qPCR.

- Include a plot of absolute levels of GFP mRNA vs RFP mRNA in each genetic condition to ensure that the mutations don't differentially affect the RFP and GFP-containing transcripts.

- Plot absolute GFP/RFP mRNA vs GFP/RFP fluorescence for each reporter and genetic background mentioned above to ensure that mRNA levels alone do not account for apparent changes in GFP/RFP fluorescence ratios.

Absolute quantification is described in this link: https://www.thermofisher.com/us/en/home/life-science/pcr/real-time-pcr/real-time-pcr-learning-center/real-time-pcr-basics/absolute-vs-relative-quantification-real-time-pcr.html)

*Reviewer #2:*

In prior work, Grayhack and colleagues identified pairs of adjacent codons that cause ribosome stalling. In the current work, they conduct an unbiased genetic screen and show that mutations in the ribosomal proteins Asc1 and Rps3, and in the protein Mbf1, promote frameshifting on these inhibitory codon pairs. Whereas Rps3 and Mbf1 appear to cooperate in maintaining reading frame, because double mutants do not enhance the phenotype of single mutants, Asc1 appears to promote frameshifting by limiting ribosome quality control pathways that turnover the stalled ribosomes (the *asc1* mutation also enhances readthrough of an in-frame reporter containing an inhibitory codon pair). Additional studies demonstrate that the nucleotides flanking the codon pairs can impact their ability to promote frameshifting. Finally, purification of the frameshifted proteins enabled the authors to show that mutation of these factors results in a +1 frameshift with the first of the codon of the bad codon pair in the P site.

While all three of these proteins were previously implicated in frameshifting by other groups (primarily Mike Culbertson), this work is independent of the previous work significantly expands the analysis to provide greater insight into the nature of the frameshift and the complementary versus independent roles of Rps3, Mbf1 and Asc1.

The function of Mbf1 is unclear. This protein has been implicated in both transcription and translation. As the assay for frameshifting involves a dual GFP-RFP reporter in which GFP and RFP are transcribed from the divergent GAL1,10 promoter, an important control is to examine RFP and GFP mRNA levels in the WT and mutant strains. While perhaps it is unlikely that the *mbf1* mutant would differentially affect transcription of the GFP and RFP reporters, analysis of these mRNA levels for the in-frame and +1 frameshift reporters would more definitively rule out this possibility.

*Reviewer #3:*

The paper by Wang et al., identified a eukaryotic-specific mechanism to prevent frameshifting by stalled ribosomes. Previous work from the Grayhack group, through the use of genetic screens, identified a role for the ribosome-associated factor Asc1/Rack1 in maintaining frame during stalling events. Here, the authors elaborate on these screens and identified the multi-bridging factor 1 (Mbf1) as an additional important player in preventing frameshifting on inhibitory codons. Additional regions on ribosomal protein Rps3 were also identified as potentially functionally important in mediating Mbf1's function. In particular, mutations of residues on the solvent phase of the protein's mRNA entry tunnel region promote frameshifting, but do not increase it in an *mbf1* background. Furthermore, the effect of these mutations is suppressed by overexpression of Mbf1. In contrast, overexpression of Mbf1 did not rescue the effect of *asc1* deletion on frameshifting. This, together with the observation that *mbf1* deletion does not promote readthrough of inhibitory codon (unlike *asc1* deletion), suggest that the Mbf1 and Asc1 work independently of each other. The authors put forward a model explaining these results: Mbf1, in combination with Rps3, prevents frameshifting by stalled ribosomes, whereas Asc1 is responsible for recruiting downstream factors to dissociate the ribosome and degrade the nascent proteins.

Overall, this is a very thorough investigation of frame maintenance during stalling and the requirement of different factors to prevent frameshifting. While many studies have looked at the effects of stalling from an mRNA- and a protein-quality-control perspective, only a few have explored its effect on ribosome function. The study offers new insights into how different factors communicate with the ribosome to trigger multiple pathways to prevent frameshifting and initiate ribosome rescue.

1) The paper would benefit from a model that could explain Mbf1's function. Do the authors propose a model whereby Mbf1 binding to the mRNA prevent it from slippage? Or is it its ribosome binding that prevent the ribosome from frameshifting? I understand testing either model is beyond the scope of the current paper, but it would be nice to see how the authors are thinking about the mechanism of Mbf1.

2) Although unnecessary, it would be nice if the authors analyzed the ribosome profiling data of the Gilbert group on *asc1* deletion to see if frameshifting occurs on the native targets that are mentioned in Figure 5—figure supplement 1B.

*Reviewer #4:*

In this manuscript the authors present the results of a screen (for growth on -Ura) conducted to look for mutants that confer an increase in +1 frameshifting when ribosomes are confronted with problematic CGA rich sequences within an ORF. Importantly, the screen was conducted in a background carrying two copies of Asc1 to avoid its (re)isolation as an interesting mutant. The screen yielded recessive mutations in Mbf1 and dominant mutations in the r-protein S3. The authors then proceeded to characterize in detail the frameshifting and readthrough properties of the various mutants, whether they act independently or synergistically, in the original growth assay as well as a great number of GFP reporter assays. The authors also include in this analysis deletions of Asc1, an r-protein previously implicated in readthrough and frameshifting on CGA-enriched sequences, and in ribosome quality control events. Based on the results in the manuscript, there is little doubt that Mbf1 and S3 play important roles in the events that take place on ribosome stalling at problematic CGA rich sequences. Disappointingly, this manuscript does not give much sense of what these factors might do at a molecular level as essentially the only experiment presented is a broad set of in vivo reporter assays that provide modest mechanistic insight. Also, importantly, given the well known connection between translational stalling and mRNA degradation, in particular as already detailed for Asc1, it seems essential that the authors take some effort to reconcile these diverse effects (while they do present data in Figure 3D, they don't acknowledge that there are substantial mRNA level effects that could explain a fraction of their +1 FS data). It is critical that this point be addressed. Overall, this manuscript extends the set of molecular players involved in specifying ribosome function at problematic mRNA sequences and provides a strong foundation for subsequent molecular characterization.

Specific points:

Not clear it is necessary to show plates and GFP reporter data in parallel throughout; results are simplest to digest in bar graphs from reporters (especially if both results are effectively the same).

Figure 3C – discussion of results with Hel2/Slh1 was confusing – my look at the data says there is nothing interesting going on with these double mutants (even if the asterisks say otherwise).

Figure 3D – mRNA levels clearly do vary – these values should be rigorously determined for CGA-4 and CGA-4-+1 reporters – the expectation is that mRNA level changes in various genetic backgrounds will be similar, but that protein output will be quite distinct – these data should be presented in a scatter plot that allows correlations in mRNA and protein levels to be directly compared.

Figure 4 – revisiting of di-codon pairs does not add much to what the authors learn in this study – the same pairs that were problematic in Cell paper appear to be problematic still, and are where the +1 FS signal is seen – supplemental?

Figure 5 – huge amount of work to define some vague contextual contributions to the +1FS phenomenon – doesn't add much in terms of mechanism or even the rules for context – supplemental?

Figure 6 – very nice set of data identifying CGA in P site as the critical feature for permitting the +1 FS (or stabilization of mRNA).

---

## [Author Response]

We have received the comments from four expert reviewers all of whom found the genetic screen and its follow-up to contain interesting new experiments focused on the activities of the ribosome in the face of elongation distress. These studies argue that a factor, Mb1, previously implicated in transcription and translation, and the ribosomal protein S3 play a critical role in preventing frameshifting at sites of stalling. In the absence of frameshifting, when severe stalling occurs, the ribosome-associated protein Asc1 triggers a diverse set of quality control events that have been studied in some substantial detail by the field. While all the reviewers felt that this study nicely leveraged genetics to reveal unanticipated players in the ribosome-centric event, all were somewhat concerned that the study did not go sufficiently beyond the reporter-based screen to reveal molecular insights into the roles of these factors on the ribosome. There was extensive use of reporters, with very similar disruptive features, to detail the subtleties in the response to pause sequences, for example. The figures detailing the di-codon pair experiments (Figure 4) and experiments focused on a natural context (Figure 5) provided little new information and potentially could be placed in a supplemental figure.

We condensed these sections and combined the Figure 4 and Figure 5, removing many of the panels to supplementary information. We re-wrote this section to make two main points. First, we found that Mbf1 and Asc1 act on the same sequences for both codon pairs and no-go decay. This observation supports the idea that their functions are strongly interconnected. Second, we identified a highly compact and efficient frameshifting site in the CGA-CGG-C sequence and found that the efficient frameshifting at this sequence is observed in other contexts (that happen to occur in native genes).

The most critical set of experiments that brought all the ideas in the manuscript together was Figure 6, where depletion and over-expression of tRNAs and mass spec were used to determine that the frameshifting is triggered only when CGA is in the P site with its associated I:A wobble base pairing.In addition to some concerns that molecular mechanism was lacking, the reviewers all had concerns about the relative lack of analysis and discussion of the role of mRNA decay in the events being studied. As you will read in the detailed comments, the reviewers felt it is critical to generate RNA quantitation data on both the GFP and RFP mRNAs, and then to present the data in terms of these measurements. It is clear that there are some substantial effects on mRNAs levels that result from the various genetic mutations, and it likely that these effects do not account for the observed increases in GFP only on the +1 constructs. To support the strength of this conclusion, the RNA levels should be quantitated both for the (CGA)4 constructs as well as the (CGA)4+1 constructs, as the RNA changes will likely be equivalent, and there will only be increases in GFP on the +1 constructs. While we are relatively confident that the conclusions will hold, given the extent to which translation and decay are coupled in these pathways, this analysis seems critical, and a careful discussion important for the reader.

We have done this, as detailed below. The analysis using protein/ mRNA levels yields the same main conclusions (as that using protein levels), but these conclusions are more convincing (and easier to justify) with the RNA analysis. In Figure 2B and Figure 3A, we present GFP/RFP protein (fluorescence), mRNA and protein/mRNA panels for strains with mutations in MBF1, ASC1, and RPS3 (as well as wild type and relevant double mutants).

Neither the *mbf1* nor the *RPS3-K108E* mutants have substantial effects on mRNA from in-frame or +1 frame reporters, while deletion of ASC1 does affect mRNA levels (as reported previously). The *asc1* mutant exhibits a 20.8-fold increase in frameshifted protein/mRNA, compared to a 1.1-fold increase in the in-frame protein /mRNA. Thus, the *asc1* mutant clearly affects frameshifting. The protein/mRNA analysis is a far superior method to evaluate the relationships between mutants. For both the *asc1Δ mbf1Δ* double mutant and the *RPS3-K108E mbf1Δ* double mutant the increase in frameshifting is small compared to that in the *mbf1Δ* mutant (1.2 and 1.1-fold). This is a strong argument that these components act in one pathway.

We used RT-qPCR and methods of quantification described by reviewer 1 to obtain these measurements.

In addition to this main criticism (related to measuring RNA levels carefully), there are some smaller points to be addressed in the reviewer comments. For example, the data and discussion on Hel2/Slh1 seemed over-interpreted or unclear. At a glance, there are no substantial differences that result when these factors are deleted.

We address these below.

Moreover, the discussion of how these pathways might intersect or what these data might mean were not easy to follow.

We have re-written the Discussion section on this point and added Figure 6A to clarify the relationship.

The manuscript would benefit from removal from the main text of figures that add little to our understanding of the role of these new factors (Mbf1 and S3) but simply support the results of the initial screen (yeast growth plates and the very detailed study in Figure 5, for example).

We have done this. We moved several panels to supplementary material: the spot test (panel E) from Figure 2; the complete analysis of frameshifting at CGA-CCG, CGA-AUA (Panels C and D) from Figure 4; the identification of the first CGA-CGG codon pair as the source of high efficiency frameshifting and the complete analysis of frameshifting at CGA-CGG (Panels A, B, D) from Figure 5.

We also removed several panels from the manuscript entirely: the analysis of *hel2* and *slh1* effects on frameshifting, and a small analysis of mRNA levels (panels C and D) from Figure 3. The complete mRNA measurements supersede the isolated measurement in Figure 3D. We removed tables from Figure 3, Figure 4 and Figure 5.

What would benefit from more discussion is a more complete analysis about how these results fit in with the literature on mRNA decay, and most importantly, some thoughts about what Mbf1 might be doing. A model figure with the relative positions of S3 and Asc1 on the ribosome might help frame such a discussion, even in the absence of new biochemical information on Mbf1.

We have completely rewritten the Discussion section to include and clarify these points. The second and third paragraphs of the Discussion section address the intersection of the Asc1 and Mbf1 pathways and the potential role of mRNA, including information from the literature on decay. In the fourth and fifth paragraphs, we address potential molecular mechanisms by which Mbf1 may act and have added a figure to illustrate two hypothesis.

Reviewer #1:[…] The identification of the Mbf1/Rsp3 system of translational frame maintenance is an important contribution to the field and opens up new areas of research. Furthermore, this work represents a systematic genetic dissection of the determinants of translational frameshifting, and the authors confirm the conclusions informed by synthetic stalling sequences using endogenous genomically-encoded sequences. However, additional and critical controls are necessary to support the authors' conclusions. Specifically, this work relies heavily on a ratiometric, fluorescent reporter system based on a bidirectional pGAL1,10 promoter. In this reporter, two separate mRNAs are created, one to use as an expression control (RFP) and one as an experimental condition that encodes various test sequences upstream of a GFP in different frames. By comparing protein expression levels between the two messages, the authors hope to quantify the effect of the experimental conditions on translation. Yet this system also is sensitive to changes that affect mRNA levels of sequences that stall ribosomes. In fact, all the results reported in this study could be the result of how Asc1, Mbf1, and Rps3 affect decay of mRNAs that encode stalls and frameshifts. This is more than a hypothetical possibility, as asc1∆ has been reported to affect mRNA levels of messages with stalls (Kuroha et al., 2010; Ikeuchi et al., 2016; Sitron et al., 2017).

Changes in GFP and RFP protein could indeed be due to changes in RNA levels. Thus, we measured GFP and RFP mRNA levels for (CGA) reporters (in-frame and out-of-frame) in the wild type strain as well as in *mbf1, RPS3, asc1*, and double mutants. The specific information is detailed below in response to individual comments.

The results reported in Figure 2B and Figure 3A were done by measuring GFP and RFP fluorescence and mRNA levels in triplicate in a single experiment. We also measured GFP and RFP fluorescence and mRNA levels in triplicate for the in frame optimal codon reporter (AGA) in Figure 2B in the wild type strain as well as in *mbf1, RPS3*, and double mutant. Similarly, we measured protein and mRNA simultaneously for the results reported in Figure 3A. As we described above, mutations in *mbf1* and *RPS3* do not significantly affect mRNA levels of in-frame or +1 reporters, while mutation of ASC1 has the expected effects on both in-frame and +1 frame mRNA levels. However, the 55-fold increase in frameshifted GFP/RFP in the *asc1* mutant (relative to wild type) is not accounted for by the 2.7-fold increase in mRNA; the increase in protein/mRNA of frameshifted GFP/RFP is 20-fold, while that of in-frame GFP/RFP is only 1.1-fold.

1) The pGAL1,10-RFP/insert-GFP reporter system needs additional validation to ensure that measurements of GFP/RFP values truly report on increases in the frequency of stall readthrough or frameshifting rather than mRNA metabolism. This validation is of particular concern due to the data presented in Figure 3D. We plotted the mRNA levels of the (CGA-CGA)3+1 reporter against the corresponding GFP/RFP fluorescence values and found there is an almost perfect linear relationship between the levels of GFP mRNA and GFP/RFP fluorescence in the wt, asc1∆, mbf1∆, and asc1∆mbf1∆ strains. The slope we plotted is not 1, however, since we do not have the standard curves for these primer sets, we cannot know the absolute differences in mRNA, only the relative increase. Thus, the increase in GFP/RFP fluorescence in the mutant strains could be driven primarily by an increase in mRNA levels of the stalling transcript rather than increased frameshifting frequency.To validate the reporters used in the context of frameshifting and stalling, the authors must measure GFP and RFP mRNA levels of (CGA-CGA)3, (CGA-CGA)3+1, (CGA-CGA)3-1 reporters using absolute quantification (not relative quantification) in wt, asc1∆, mbf1∆, and asc1∆mbf1∆ strains by RT-qPCR.

We have done this. We used the absolute standard curve method described below to quantify these mRNAs. We cloned GFP and RFP sequences into plasmid pSP73, transcribed restriction cut plasmids with T7 RNA polymerase, digested DNA and removed small molecules in a spin column (using the RiboMAX Large Scale RNA Production System-T7 from Promega).

- Include a standard curve for each primer set (varying cDNA input) to map experimental conditions back to absolute input levels. Ensure the assay is conducted in the linear range of the qPCR.

We generated standard curves with each primer set using in vitro transcripts of a cloned GFP and RFP regions. We demonstrated that the in vitro transcribed RNA was intact and a single species. Each set of RT-qPCR reactions contained the 5 point standard curves and all measured values fell within the linear range.

- Include a plot of absolute levels of GFP mRNA vs RFP mRNA in each genetic condition to ensure that the mutations don't differentially affect the RFP and GFP-containing transcripts.

We have included plots of GFP mRNA vs RFP mRNA for in-frame and out-of-frame reporters in each genetic background in Figure 2—figure supplement 1A and in Figure 3—figure supplement 1C.

- Plot absolute GFP/RFP mRNA vs GFP/RFP fluorescence for each reporter and genetic background mentioned above to ensure that mRNA levels alone do not account for apparent changes in GFP/RFP fluorescence ratios.

We have plotted GFP/RFP fluorescence versus GFP/RFP mRNA in Figure 2—figure supplement 1B and in Figure 3—figure supplement 1D. We have also reported GFP/RFP fluorescence normalized by GFP/RFP mRNA in Figure 2B and 3A. There are only small differences in mRNA levels for *RPS3* and *mbf1* mutants in Figure 2. By contrast, mutation of ASC1 does affect mRNA levels. As shown in Figure 3—figure supplement 1D, the changes in mRNA levels correlate with in-frame expression of GFP/RFP, and with the increase in frameshifted GFP/RFP in the *asc1Δ mbf1Δ*, compared to the *mbf1Δ* strain. However, this same analysis indicates that the increase in mRNA in the *asc1* mutant relative to the wild type does not correlate with the increase in frameshifted GFP/RFP.

Absolute quantification is described in this link: https://www.thermofisher.com/us/en/home/life-science/pcr/real-time-pcr/real-time-pcr-learning-center/real-time-pcr-basics/absolute-vs-relative-quantification-real-time-pcr.html)

We thank the reviewer. This methodology helped us.

Reviewer #2:[…] The function of Mbf1 is unclear. This protein has been implicated in both transcription and translation. As the assay for frameshifting involves a dual GFP-RFP reporter in which GFP and RFP are transcribed from the divergent GAL1,10 promoter, an important control is to examine RFP and GFP mRNA levels in the WT and mutant strains. While perhaps it is unlikely that the mbf1 mutant would differentially affect transcription of the GFP and RFP reporters, analysis of these mRNA levels for the in-frame and +1 frameshift reporters would more definitively rule out this possibility.

We did this as explained above.

Reviewer #3:[…] 1) The paper would benefit from a model that could explain Mbf1's function. Do the authors propose a model whereby Mbf1 binding to the mRNA prevent it from slippage? Or is it its ribosome binding that prevent the ribosome from frameshifting? I understand testing either model is beyond the scope of the current paper, but it would be nice to see how the authors are thinking about the mechanism of Mbf1.

We appreciate the reviewer’s suggestion. In fact, we proposed two models, both of which depend upon the interaction of Mbf1 with both mRNA and the ribosome (RPS3). The first posits that this interaction occurs between Mbf1 and the leading ribosome, while the second posits that Mbf1 interacts with colliding ribosomes (perhaps interacting with both Rps3 and Asc1). These models are presented in Figure 6B and are discussed in the Discussion section.

New Text:

“There are two reasonable models to account for the role of Mbf1 and Rps3 in reading frame maintenance (Figure 6B). The first model is that Mbf1 has a loose association with mRNA and is recruited to the leading stalled ribosome by an interaction with Rps3; the interactions with the ribosome and the mRNA at the stall site could restrict mRNA movement in the ribosome. Based on structures of prokaryotic ribosomes caught in translocation, mRNA flexibility may occur in ribosomes lacking an A site tRNA due to few contacts with the region of mRNA near the A site (Zhou et al., 2013), or due to a failure of two rRNA pawls that lock the mRNA in a translocating ribosome (Zhou et al., 2013), or due to defects in the interactions with elongation factor 2 (Zhou et al., 2014). The ribosome stall might allow sufficient time or altered structure that one of these occurs. The second model, which is based on the observation that ribosome collisions trigger no-go decay (Simms et al., 2017b), is that Mbf1 is recruited to colliding ribosomes to buffer the collision effects; in this case Asc1 and Rps3 might both participate in Mbf1 recruitment. Mbf1 could prevent ribosome collision-mediated movement of the leading ribosome on the mRNA.”

2) Although unnecessary, it would be nice if the authors analyzed the ribosome profiling data of the Gilbert group on asc1 deletion to see if frameshifting occurs on the native targets that are mentioned in Figure 5—figure supplement 1 B.

Gilbert’s lab generously supplied us with all the ribosome reads mapped to the genome. We attempted to examine frameshifting by generating a histogram of ribosome reads for each nucleotide in the AYT1 gene (in the *asc1* mutant). However, there were very few ribosome footprints downstream or around the CGA-CGG region in the AYT1 gene.

Reviewer #4:[…] Specific points:Not clear it is necessary to show plates and GFP reporter data in parallel throughout; results are simplest to digest in bar graphs from reporters (especially if both results are effectively the same).

We have moved the spot test data (Figure 2E) to the supplement.

Figure 3C – discussion of results with Hel2/Slh1 was confusing – my look at the data says there is nothing interesting going on with these double mutants (even if the asterisks say otherwise).

The reviewer is correct; we removed the analysis of *Hel2* and *Slh1* entirely.

Figure 3D – mRNA levels clearly do vary – these values should be rigorously determined for CGA-4 and CGA-4-+1 reporters – the expectation is that mRNA level changes in various genetic backgrounds will be similar, but that protein output will be quite distinct – these data should be presented in a scatter plot that allows correlations in mRNA and protein levels to be directly compared.

We have done this and have presented protein per mRNA levels in Figure 2B and 3A.

Figure 4 – revisiting of di-codon pairs does not add much to what the authors learn in this study – the same pairs that were problematic in Cell paper appear to be problematic still, and are where the +1 FS signal is seen – supplemental?

We examined frameshifting at the 12 most inhibitory codon pairs to find out if Mbf1 and Asc1 acted on the same sequences. We think this analysis provided evidence that Asc1 and Mbf1 generally act together. We found that very efficient frameshifting only occurred at three pairs, and these were the same three pairs at which the deletion of ASC1 suppressed the expression defect. We did remove the complete analysis of both in-frame and frameshifting in all strains to the supplement, since it did not add conceptually to the manuscript.

Figure 5 – huge amount of work to define some vague contextual contributions to the +1FS phenomenon – doesn't add much in terms of mechanism or even the rules for context – supplemental?

The purpose of this analysis was to identify an efficient and localized frameshifting site. To streamline the manuscript, we moved the material from Figure 5A and 5B to the supplement. We also summarized the result from these figures in a single sentence. Subsection “Mbf1 and Asc1 work at a common subset of inhibitory codon pairs and at a single 304 inhibitory codon pair in a context-dependent manner”: “To define the source of efficient frameshifting in the CGA-CGG reporter, which has 3 CGA-CGG pairs (Figure 4A), we initially determined that the first CGA-CGG codon pair was responsible for highly efficient frameshifting (Figure 4D, Figure 4—figure supplement 2A and B).”

The section on native sequences is also shortened.

Figure 6 – very nice set of data identifying CGA in P site as the critical feature for permitting the +1 FS (or stabilization of mRNA).

Thank you.